

# Assessing the Impact of Earth Observation Data-Driven Calibration of the Melting Coefficient on the LISFLOOD Snow Module

Valentina Premier[1,*], Francesca Moschini[2,4,*], Jesús Casado-Rodríguez[2], Davide Bavera[3], Carlo Marin[1], and Alberto Pistocchi[2]

[1]Institute for Earth Observation, Eurac Research, Bolzano, Italy.
[2]European Commission, Joint Research Centre (JRC), Ispra, Italy.
[3]Arcadia SIT, Milano, Italy.
[4]Rey Juan Carlos University, Madrid, Spain.
[*]These authors contributed equally to this work.

**Correspondence:** Valentina Premier (valentina.premier@eurac.edu) and Francesca Moschini
(francesca.moschini@ec.europa.eu)

**Abstract.** LISFLOOD is a comprehensive hydrological model widely used in Europe. Among various hydrological processes, it simulates snowmelt using a degree-day-based snow module. Traditionally, the snowmelt coefficient is calibrated using discharge data. This study evaluates LISFLOOD's current snow module and explores an alternative calibration approach based on Earth Observation (EO) derived snow cover fraction (SCF) observations across nine European basins with varying degrees

of snow cover influence. We utilize a novel integration of Sentinel-2 and MODIS data to address issues related to data gaps and missed snow cover detection in complex topography. Using EO SCF, we estimate a spatially distributed snowmelt coefficient, which contrasts with the uniform coefficients currently used in LISFLOOD. The new calibration approach, involving an optimization routine to match modeled and observed SCF, outperforms a previous method that did not deal with fractional snow as well as discontinuous snow cover periods. When compared with EO SCF, the traditional calibration shows bias values

ranging from -0.56% to 22.50%, with root mean squared error (RMSE) values varying from 20.43% to 54.64%. We obtained improvements up to 8% both in bias and RMSE when the optimization approach is used. While the optimized coefficients did not significantly alter simulated discharge, the improved SCA representation led in some cases to shifts in the timing and magnitude of snowmelt and total runoff. These findings highlight the potential of integrating EO data to enhance snowmelt simulations and improve water balance predictions, with important implications for hydrological modeling and water resource

management.

## 1 Introduction

Snow cover plays a crucial role in hydrological processes, particularly in snow-fed regions, where snowmelt significantly contributes to river runoff. In mountainous regions and snow-dominated basins, snowmelt often represents a major source of



river discharge. Depending on geography and climate, the contribution of snow to river runoff can vary substantially, from as low as 40% up to 95% of the total annual flow (Viviroli et al., 2007). This makes accurate modeling of snowmelt processes essential for both flood forecasting and drought monitoring and prediction, especially in snow-dominated areas and in the context of a changing climate (Barnett et al., 2005; Blöschl et al., 2017; Beniston et al., 2011). Hence, the snow module in hydrological models must be carefully designed and assessed for a correct management of water resources.

LISFLOOD is one of the most comprehensive operational models used in Europe to simulate hydrological processes (De Roo et al., 2000; Van Der Knijff et al., 2010; Burek et al., 2013). Developed by the European Commission's Joint Research Centre (JRC) for the Copernicus Emergency Management Service (CEMS), it is an integrated, distributed hydrological model that underpins both the European Flood Awareness System (EFAS) (JRC, 2024a; Matthews et al., 2025b) and the Global Flood Awareness System (GloFAS) (JRC, 2024b; Matthews et al., 2025a). LISFLOOD also supports the European and Global

Drought Observatories (EDO and GDO) by providing key indicators (see https://drought.emergency.copernicus.eu/) (Cammalleri et al., 2015, 2017). Primarily used for operational flood forecasting across Europe, LISFLOOD simulates several hydrological processes, including surface runoff, infiltration, groundwater recharge, and snowmelt. The snow module within LISFLOOD simulates the snowmelt through a temperature-based approach, specifically a degree-day model (Martinec and Rango, 1981). These relatively simple but robust models are commonly employed in large-scale hydrological models. The ac-

curacy of the snow module is crucial for regions where snowmelt significantly impacts river flow. In most hydrological models, including LISFLOOD, the snow module is typically calibrated using discharge observation data. However, discharge data alone may not provide sufficient information to accurately identify all parameter values, particularly when dealing with distributed processes (Beven, 2012). On the other hand, calibration against spatially distributed, satellite-based snow cover data offers significant advantages in overcoming this limitation (Franz and Karsten, 2013). Snow cover area (SCA) from satellites has proven

to be very accurate (e.g., Tedesco, 2015; Bormann et al., 2018). Nevertheless, the coarse spatial resolution and the frequent cloud obstruction may limit detailed assessments, especially in complex and forested terrains (Riggs et al., 2015; Engel et al., 2017). By directly estimating the snowmelt coefficient from snow cover information, the calibration of other model parameters is simplified (Asaoka and Kominami, 2013; Riboust et al., 2019; Gyawali and Bárdossy, 2022; Ruelland, 2024). In the past, low-resolution products such as MODIS, which provides SCA at 500 m resolution, have been used for extensive comparisons

of the LISFLOOD snow module (Thirel et al., 2012; Pistocchi et al., 2017). While the results have shown reasonable accuracy in validating snow models when considering SCA aggregated at the basin level, detailed analyses at the pixel level are lacking, especially when pixels present fractional snow cover during the melting period. Therefore, pixel-wise performances in terms of snow cover fraction (SCF) should be assessed.

The aim of this work is to explore the capabilities and limitations of the snow module within the LISFLOOD model in
reproducing realistic snow cover distribution and accurate discharges across Europe. To accomplish this, we first assess the performance of the LISFLOOD snow module using a novel remote sensing snow cover product with a daily temporal resolution and 50 m spatial resolution. The SCA dataset is derived by fusing Sentinel-2 and MODIS data through gap-filling and downscaling techniques, which have shown high accuracy (Premier et al., 2021). Starting from a high-resolution (HR) product preserves spatial detail even after aggregation, which is crucial for accurate modeling and analysis. A detailed evaluation of this





dataset against other existing snow cover products highlighted its key strengths, particularly the availability of a fully gap-filled

daily dataset. This novel dataset serves as a benchmark for evaluating the snow module's performances across Europe.

Secondly, this paper focuses on the snowmelt coefficient calibration — a key parameter within the simplified snowmelt model of LISFLOOD. We propose a dedicated calibration of the coefficient by using spatially distributed snow cover information derived from earth observation (EO) data as a benchmark. This calibration approach differs from traditional hydrolog-

ical calibration methods by introducing an independent process that does not rely solely on discharge data. Furthermore, the snowmelt coefficient is estimated on a pixel-by-pixel basis, enhancing the accuracy of snowmelt representation across different landscapes. This represents an important improvement over the LISFLOOD approach, which results in a lumped coefficient, constant for each subcatchment. This study proposes to calibrate the snowmelt coefficient using: i) a previously proposed method based on the number of snow-covered days per pixel for estimating the coefficient (Pistocchi et al., 2017), and ii) a

novel method that minimizes the error between LISFLOOD and EO SCF. The second is possible thanks to the introduction of i) an appropriate parameterization to convert SWE to SCF, and ii) the use of a daily gap-filled SCF time-series based on HR EO snow cover data.

The study sites encompass a variety of watersheds selected to represent diverse topographical and climatic conditions across Europe, including major mountain ranges and a wide range of latitudes. These basins also feature different land cover types,

from rugged mountainous terrain to flat, forested areas, offering a comprehensive foundation for evaluating the model's performance in diverse environmental contexts.

## 2  Material and Methods

### 2.1  Daily Snow Cover Area Retrieval

Given the trade-off between spatial and temporal resolution in current multi-resolution satellite missions, there is a shortage of

daily high-resolution (HR) optical data necessary for effective snow cover change monitoring. Low-resolution (LR) sensors, such as MODIS and VIIRS, provide daily SCF data, but their coarse spatial resolution limits their effectiveness in complex alpine and heterogeneous environments, where finer detail is essential (Molotch and Margulis, 2008). In contrast, HR multi-spectral sensors, like Sentinel-2, offer more spatially detailed snow cover data, but their lower revisit frequency, due to orbital constraints, reduces their utility for continuous monitoring of rapidly changing snow conditions.

Additionally, all optical sensors face challenges due to cloud cover, which can obscure the ground for extended periods and hinder snow detection. This is particularly problematic in mountainous regions, where snow patterns are highly variable (Parajka and Blöschl, 2006). Currently available operational products struggle to overcome these limitations, leading to significant data gaps. To address this issue, we integrate multi-resolution optical satellite data to obtain daily HR SCA information, specifically binary data indicating snow presence. We apply the approach presented by Premier et al. (2021) to fill temporal

and spatial gaps. The approach to creating a continuous, gap-filled daily SCA dataset relies on two primary data sources, i.e., i) a LR SCF product based on optical data, as MODIS data, and ii) HR optical data, particularly Sentinel-2 snow maps. The methodology builds on the intuitive idea that inter-annual spatial patterns are influenced by local topography and meteorologi-



cal conditions. Areas with similar elevation, slope, and aspect tend to exhibit comparable responses to snow processes, such as accumulation, distribution, and melting. A long time series of HR acquisitions is required to learn the recurring patterns. The workflow leverages this concept through an iterative gap-filling and downscaling procedure, followed by a machine learning step. Importantly, the approach is designed to be independent of the HR or LR input snow cover product. In this study, our focus will be on operationally available datasets with the main aim to broaden the potential use of this approach. For more details on the daily SCA retrieval approach, refer to Premier et al. (2021).

Once the HR daily gap-filled SCA binary snow cover is derived, it is aggregated and resampled to the grid of the LISFLOOD model resulting in an SCF value for each pixel. We refer to this time-series as EO-SCA. Appendix A details the HR and LR snow cover data used to retrieve daily snow cover in this study, and provides further insights into the use of alternative gap-filled snow cover datasets that are operationally available.

## 2.2 Snow Module of LISFLOOD

LISFLOOD is an open-source spatially distributed hydrological model designed to simulate hydrological processes in large European river basins. The model simulates the whole water cycle and comprises multiple modules to reproduce various hydrological processes such as surface runoff, infiltration, groundwater recharge, and snowmelt. A complete description of the model is provided by De Roo et al. (2000); Van Der Knijff et al. (2010); Burek et al. (2013). The current model setup operates on a 1 arc-minute grid resolution (approximately 1.4 km), covering the entire European Union, the European continent, and the Mediterranean coast. It can be applied at multiple scales, from large river basins to global regions, supporting flood forecasting, water resource assessments, and analysis of factors like water demand, river regulation, land-use changes, and climate change. Precipitation and temperature fields used as input for the model are from the EMO-1 dataset (Thiemig et al., 2022), used as forcing in the EFAS. At the moment, the model is the core component of the European operational flood and drought monitoring and forecasting system within the CEMS. The same applies at a global scale (3 arc-minute resolution) for GloFAS, EDO and GDO.

The model is calibrated on 14 parameters, including the snowmelt coefficient, using the Distributed Evolutionary Algorithms in Python (DEAP) (Fortin et al., 2012). The optimization is based on the modified Kling-Gupta Efficiency (KGE) (Kling et al., 2012) objective function, calculated from simulated and observed river discharge.

$$KGE = 1 - \sqrt{(r-1)^2 + (\beta-1)^2 + (\gamma-1)^2} \qquad (1)$$

where $\beta$ is the bias ratio (Eq. 2), $\gamma$ is the variability ratio (Eq. 3) and $r$ is the Pearson correlation.

$$\beta = \mu_s/\mu_o \qquad (2)$$

$$\gamma = \frac{\sigma_s/\mu_s}{\sigma_o/\mu_o} \qquad (3)$$





$\mu_s$ and $\mu_o$ are simulation and observation mean, $\sigma_s$ and $\sigma_o$ are simulation and observation standard deviation.

When multiple stations are available within a catchment, the calibration process employs a cascading approach. The basin is partitioned into inter-catchments, and each inter-catchment is calibrated sequentially from upstream to downstream. The parameters for ungauged basins are estimated using a regionalization approach (Beck et al., 2016), whereas coastal and endorheic catchments (area below 150 km$^2$) use default parameters. Note that the calibration assigns a single value to each parameter across all pixels within an inter-catchment.

For the purpose of this study, we will focus exclusively on the snow module and, in detail, on the snowmelt coefficient. In LISFLOOD, the SWE is calculated by accounting for both the accumulation and melting processes:

$$\text{SWE} = P_{snow} - M \tag{4}$$

where $P_{snow}$ is the solid precipitation and $M$ is the snow melt, both of them expressed in $mm$. The total precipitation is split in solid precipitation $P_{snow}$ and rain $R$ based on a temperature threshold that is set as $1°C$. The melting is given by the following equation:

$$M = \begin{cases} (C_m + C_{seas}) \cdot (1 + 0.01 \cdot R \cdot \Delta t) \cdot (T_{avg} - T_m) \cdot \Delta t, & \text{if } T_{avg} > T_m \\ 0, & \text{if } T_{avg} \leq T_m \end{cases} \tag{5}$$

where $C_m[\frac{mm}{°C \cdot day}]$ is the snowmelt coefficient or degree-day factor, $C_{seas}[\frac{mm}{°C \cdot day}]$ is a degree-day factor introduced to account for seasonal effects, $R[\frac{mm}{day}]$ is the rainfall intensity, $T_{avg}[°C]$ is the average daily temperature, $T_m[°C]$ is the temperature threshold at which snowmelt occurs — set as $1°C$— and $\Delta t[days]$ is the time interval — set as 1 day. The degree-day factor for the seasonal effects is computed as follows:

$$C_{seas} = \frac{1}{2} \sin\left(\frac{2\pi}{365}(\text{doy} - 81)\right) \tag{6}$$

Note that the sum $C_m + C_{seas}$ must remain positive. Since $C_{seas}$ fluctuates between -0.5 and 0.5, $C_m$ must be at least 0.5 to ensure this condition is satisfied. Furthermore, snow melt and accumulation are modeled separately for 3 separate elevation zones to take into account sub-pixel heterogeneity linked to elevation differences given the large pixel size. The model code and documentation are available at https://github.com/ec-jrc/lisflood-code/.

### 2.3 Snow Cover Parametrization

To compare the SWE output of the LISFLOOD model with the EO-SCA, it is necessary to convert SWE into SCF. This conversion implies the use of a parametrization that accounts for changes in snow depth and density within a pixel. Due to the large intra-pixel variability, the relation is influenced by factors such as topography and land cover (Roesch et al., 2001). Several parametrizations are available in the literature (Lee et al., 2024). In this study, we adopted the approach proposed by





Swenson and Lawrence (2012) used in the Community Land Model (CLM). However, the approach of Zaitchik and Rodell
(2009) was also tested and the results can be found in Appendix B.

Swenson and Lawrence (2012) differentiates the accumulation and ablation periods. The updated snow cover fraction
$SCF^{n+1}$ is given by the following equation for accumulation

$$SCF^{n+1} = 1 - [(1 - \tanh(k_{accum}\Delta SWE))(1 - SCF^n)] \tag{7}$$

where $k_{accum}$ is a constant with a default value of 0.1, $\Delta SWE$ is the amount of new SWE in mm and $SCF^n$ is the snow
cover fraction from the previous time step. Although many state-of-the-art models assume a constant value for $k_{accum}$, our
results indicate that this parameter significantly influences the outcome and likely varies pixel-wise due to topographic differ-
ences. $k_{accum}$ can be estimated by measuring SCF and $\Delta SWE$ when precipitation occurs over an initially snow-free area, as
suggested by Swenson and Lawrence (2012). To identify such a condition, we selected the first day on which both EO-SCA
and LISFLOOD simulation indicate snow presence. If a pixel never exhibits snow, the default value is retained. This process
was repeated for all available seasons, and an averaged value over time was then considered.

On the other hand, SCF is given by the following equation for melting

$$SCF = 1 - \left[\frac{1}{\pi}\arccos(2\frac{SWE}{SWE_{max}} - 1)\right]^{N_{melt}} \tag{8}$$

where $SWE_{max}$ is the threshold SWE above which SCF is 100% - it depends on the topography and the forest coverage of the
pixel - and $N_{melt}$ is a parameter that depends on the topographic variability within the grid cell and is calculated as follows

$$N_{melt} = \frac{200}{\max(10, \sigma_{topo})} \tag{9}$$

with $\sigma_{topo}$ being standard deviation of the elevation within a grid cell calculated from the Multi-Error-Removed Improved-
Terrain (MERIT) DEM with a spatial resolution of 90 m (Yamazaki et al., 2017). The threshold $SWE_{max}$ is updated when
accumulation happens as follows

$$SWE_{max} = SWE\left[\frac{\cos(\pi(1 - SCF)^{1/N_{melt}}) + 1}{2}\right] \tag{10}$$

## 2.4  Snowmelt Coefficient Estimation from EO Data

The snowmelt coefficient is estimated following two different approaches. First, the approach presented by Pistocchi et al.
(2017). Under the hypothesis of a period of continuous snow cover, we can write the following balance of snowfall and
snowmelt:

$$\sum_{i=1}^{n}(P_{snow,i} - M_i) = 0 \tag{11}$$



where $n$ is the number of days composing the continuous snow cover period. By substituting with Eq. 5, we can derive the snowmelt coefficient

$$C_m = \frac{\sum_{i=1}^{n} P_{snow,i} - \sum_{i=1}^{n} C_{seas}(1 + 0.01 \cdot R_i \cdot \Delta t)(T_{avg,i} - T_m) \cdot \Delta t}{\sum_{i=1}^{n}(1 + 0.01 \cdot R_i \cdot \Delta t)(T_{avg,i} - T_m) \cdot \Delta t} \tag{12}$$

The periods of continuous snow cover are detected from EO-SCA. While Eq. 11 may not strictly apply to pixels experiencing multiple snow cover/snow-free cycles (common at lower altitudes or temperate climates), we simplify the analysis by assuming its validity for all snow-covered days. This simplification also helps mitigate residual errors from inaccurate SCA reconstruction.

An alternative approach for estimating the melting coefficient involves formulating an optimization problem aimed at minimizing the error between LISFLOOD and EO-SCF. Specifically, we minimize the mean squared error (MSE) over time to identify the optimal coefficient that results in the smallest error. Hence, we solved the following minimization problem for each pixel:

$$\hat{C_m} = \arg\min_{C_m} \sum_{t} (\text{L-SCF}(t, C_m) - \text{EO-SCF}(t))^2 \tag{13}$$

where $\hat{C_m}$ is the optimized snowmelt coefficient and L-SCF is calculated from SWE through Eq. 4 and by applying the parametrization (Eqs. 7 and 8). To solve this optimization problem, we utilize the L-BFGS-B algorithm - a limited memory quasi-Newton, gradient-based optimization algorithm - provided in the SciPy library (Virtanen et al., 2020).

The snowmelt coefficient is estimated in both cases on a pixel-wise basis for each hydrological season. Estimated values are constrained between 0.5 and 10, as values outside this range lack physical meaning and may result from errors, such as having too few days to perform a reliable optimization(e.g., ephemeral snow at low altitudes). Specifically, according to Eq. 5 values of $C_m$ would result in negative melting, which is not physically realistic. Then, a mean value for each pixel is calculated using data from all seasons except the final one, which is used exclusively for a further independent assessment of the results. For pixels where the coefficient could not be estimated - such as those without snow during the year or where pixels were masked out due to the presence of water bodies - the original LISFLOOD coefficient is retained.

## 3 Test sites

In this study, we analysed 9 snow-dominated hydrological basins across Europe, located in Italy, Switzerland, Austria, Germany, France, Spain, Slovakia, and Sweden. These basins were selected for their strong snow influence on hydrological processes, with snow cover persisting for a significant portion of the year, and/or their particular climatic and morphological characteristics in relationship with snow processes. Representing a range of geographical contexts, these basins encompass prominent mountainous regions like the Alps and Pyrenees, as well as flatter terrains such as those in Scandinavia. Additionally, these basins vary in land cover characteristics, including different proportions of forested areas, which impact both snow



accumulation and melt dynamics. For example, Mörrumsån in Sweden and Laborec in Slovakia are flat regions with brief
and intermittent snow periods. Laborec also exhibits significant forest cover. Umeälven in Sweden, though also relatively flat,
experiences prolonged snow cover lasting several months due to its high latitude. In contrast, the remaining basins display a
more typical alpine snowpack. Of the 9 catchments, 7 were originally calibrated against observed river discharge, whereas 2,
the Guadalfeo and the Adige were parameterized using the regionalization approach.

The availability of hydrological data was a constraint for the basin selection to ensure a complete analysis. Key information
on these basins is presented in Table 1 and Fig. 1. The EO dataset covers six hydrological seasons, from October 1, 2017, to
September 30, 2023, i.e., the period with maximum availability of Sentinel-2 data (both Sentinel-2A and Sentinel-2B). The
LISFLOOD evaluation (see Sec. 4.3) covers a longer period (1992-2022) allowing to build a climatology. The grid of the model
with a spatial resolution of around 1 arc-minute (approximately 1.4 km) is considered for the analysis, as mentioned previously
in Sec. 2.2.

**Table 1.** Overview of the nine hydrological catchments selected in this study, including their respective countries, area, and elevation information.

| Basin | Countries | Area [km$^2$] | Elevation [m] | | |
| :---: | :---: | :---: | :---: | :---: | :---: |
| | | | min | mean | max |
| Adige | Italy | 12100 | 4 | 1483 | 3511 |
| Alpenrhein | Switzerland/Austria | 7400 | 406 | 1511 | 2983 |
| Arve | France | 2000 | 401 | 1343 | 3700 |
| Gállego | Spain | 3900 | 207 | 791 | 2738 |
| Guadalfeo | Spain | 1200 | 177 | 1251 | 2872 |
| Laborec | Slovakia | 1300 | 167 | 408 | 868 |
| Mörrumsån | Sweden | 3500 | 34 | 184 | 301 |
| Salzach | Germany/Austria | 6700 | 372 | 1236 | 3050 |
| Umeälven | Sweden | 6100 | 334 | 709 | 1482 |



**Figure 1.** Overview of the nine hydrological basins chosen for this study: Adige (Italy), Alpenrhein (Switzerland/Austria), Arve (France), Gállego and Guadalfeo (Spain), Laborec (Slovakia), Mörrumsån (Sweden), Salzach (Germany/Austria) and Umeälven (Sweden).





## 4 Results

### 4.1 Snowmelt Coefficient Calibration

In this section, the results of a dedicated calibration of the snowmelt coefficient $C_m$ using EO data, as described in Sec. 2.4, are reported. To differentiate between the snowmelt coefficient derived from traditional hydrological calibration and that obtained through EO data, we will refer to them as LISFLOOD $C_m$ (L-$C_m$) and earth observation $C_m$ (EO-$C_m$). Moreover, we also conducted a comparison of the coefficients obtained using the formula in Eq. 12 with those derived through the L-BFGS-B optimization. We refer to EO-$C_{m,1}$ and EO-$C_{m,2}$ to differentiate between the two approaches. Note that EO-$C_{m,1}$ and EO-$C_{m,2}$ are obtained as average values considering the first five hydrological years, while season 2022/23 is used as an independent dataset for evaluation purposes only.

In Figs. 2 and 3, the snowmelt coefficients L-$C_m$ are shown together with the new coefficients EO-$C_{m,1}$ and EO-$C_{m,2}$. The corresponding histograms are also illustrated. Missing values (in white color) correspond to masked areas, such as water bodies, or pixels that never experienced snow for the analysed period. Notable differences are evident between the three approaches, both in terms of spatial patterns and magnitude of the coefficients. Especially EO-$C_{m,2}$ shows very high values for basins or areas where ephemeral snow is present. Note that the parameter range in the LISFLOOD calibration is narrower (2.5-6.5) than in the EO calibrations. This larger range allows us to evaluate the potential of the new calibration to correct for erroneous precipitation inputs and better represent the melting phase (see next section).





**Figure 2.** Snowmelt coefficients estimated using the hydrological calibration of LISFLOOD (L-$C_m$, on the left), EO data via Eq. 12 (EO-$C_{m,1}$, in the middle), and EO data through the optimization approach (EO-$C_{m,2}$, on the right). The corresponding histograms are also included. Missing values (in white) correspond to masked areas, such as water bodies, or pixels that never experienced snow for the analysed period.





**Figure 3.** Snowmelt coefficients estimated using the hydrological calibration of LISFLOOD (L-$C_m$, on the left), EO data via Eq. 12 (EO-$C_{m,1}$, in the middle), and EO data through the optimization approach (EO-$C_{m,2}$, on the right). The corresponding histograms are also included. Missing values (in white) correspond to masked areas, such as water bodies, or pixels that never experienced snow for the analysed period.

## 4.2 Snow Cover Evaluation

To test the newly estimated coefficients, we substituted them into Eq. 5 to calculate the updated SWE. Subsequently, we compared the EO-SCA with the SCA derived from the LISFLOOD model (L-SCA), considering the different snowmelt coefficients L-$C_m$, EO-$C_{m,1}$ and EO-$C_{m,2}$.





In Fig. 4, the SCA trends with different $C_m$ are reported. Similarly, yearly scatterplots are provided in Fig. C1 in Appendix C. These results offer a quick overview of the performances at basin level. Furthermore, to assess improvements at the pixel-scale we report the metrics for SCF, using EO-SCF as the benchmark. These are computed per pixel along the temporal dimension and then averaged spatially. To specifically assess the model's ability to detect snow cover, we exclude pixels where both the target and reference SCF are 0 (snow-free). While including these pixels would lead to better performances, the results might

be biased especially when a large portion of the basin is snow-free.

In Table 2, we present the average metrics calculated across all analysed hydrological seasons for brevity. Similar metrics with a different SCF parametrization are reported in Table B1. Detailed metrics for each season individually are provided in Table C1 in Appendix C.







**Figure 4.** SCA trends for the nine river basins. In black, the EO-SCA, in light blue the LISFLOOD derived SCA calculated with L-$C_m$, in green the LISFLOOD SCA calculated with EO-$C_{m,1}$, and in orange the LISFLOOD derived SCA calculated with EO-$C_{m,2}$.



**Table 2.** Evaluation of LISFLOOD SCF in terms of BIAS, RMSE, and correlation using the EO-SCF as the benchmark. The three different snowmelt coefficients L-$C_m$, EO-$C_{m,1}$ and EO-$C_{m,2}$ were utilized. Pixels where both the target and reference SCF are snow-free are excluded from the analysis.

| | L-$C_m$ | | | EO-$C_{m,1}$ | | | EO-$C_{m,2}$ | | |
|---|---|---|---|---|---|---|---|---|---|
| | Bias [%] | RMSE [%] | $\rho$ [-] | Bias [%] | RMSE [%] | $\rho$ [-] | Bias [%] | RMSE [%] | $\rho$ [-] |
| Adige | 10.00 | 35.82 | 0.60 | 18.57 | 35.31 | 0.61 | 8.36 | 32.48 | 0.62 |
| Alpenrhein | −12.59 | 32.61 | 0.69 | 2.42 | 24.94 | 0.72 | −4.30 | 24.23 | 0.75 |
| Arve | −11.61 | 34.42 | 0.63 | 5.50 | 30.72 | 0.63 | −5.48 | 29.03 | 0.64 |
| Gállego | −0.56 | 44.46 | 0.52 | 1.46 | 43.82 | 0.54 | −1.67 | 41.01 | 0.55 |
| Guadalfeo | 7.66 | 28.46 | 0.32 | 17.95 | 30.09 | 0.34 | 6.29 | 27.27 | 0.30 |
| Laborec | 22.50 | 48.88 | 0.45 | 25.82 | 50.39 | 0.44 | 14.69 | 46.43 | 0.46 |
| Mörrumsån | 6.97 | 54.64 | 0.27 | 12.66 | 54.86 | 0.28 | 5.30 | 54.55 | 0.27 |
| Salzach | −6.83 | 36.84 | 0.67 | 7.65 | 30.11 | 0.68 | 0.18 | 28.95 | 0.69 |
| Umeälven | −1.06 | 20.43 | 0.58 | 1.72 | 17.57 | 0.64 | −0.11 | 17.72 | 0.64 |

Due to variations in the size and location of these basins, the SCA varies significantly and exhibits seasonal fluctuations
influenced by whether the year was wetter or drier. For example, while most basins reach nearly 100% snow cover at least briefly, others, such as Gállego and Guadalfeo, show very little snow cover. A generally good agreement can be observed between EO and LISFLOOD SCA, even before applying the new coefficient. However, important differences are evident in certain basins, including the Adige, Laborec, and Mörrumsån. In the Adige river basin, the LISFLOOD model shows a consistently larger SCA compared to satellite observations. These discrepancies, particularly noticeable during accumulation
phases, such as in the 2020/21 season, may stem from an overestimation of snowfall and, consequently, precipitation fields over the basin, or from an inaccurate partition of solid/liquid precipitation. Despite these differences in magnitude, the timing of SCA variations appears to align well between the two datasets. Interestingly, the Arve basin demonstrates that the SCA remains greater than 0 even during the summer season, as some pixels persist as snow or correspond to glacierized areas. Notably, in the current setup of LISFLOOD, these pixels do not persist as snow, whereas correcting the coefficient results
in a better representation. On the other hand, the differences observed in the Laborec and Mörrumsån basins are more likely attributable to challenges in snow detection by the satellite. These include prolonged periods of missing acquisitions due to cloud cover, combined with ephemeral snowfalls in flat areas, as is the case in Mörrumsån. Additionally, significant forest coverage, particularly in the Laborec basin, further complicates accurate snow detection.

In general, all three approaches yield satisfactory results when evaluated at basin scale. However, important differences are
highlighted when considering SCF (Table 2). Before dedicated EO-based calibration, the agreement is acceptable, with the highest bias observed at approximately 20% for the Laborec basin. The highest root mean square error (RMSE) values are observed for Mörrumsån and Laborec, due to the reasons previously discussed. Additionally, Gállego also exhibits notable discrepancies, likely because most pixels remain snow-covered for only short periods complicating the analysis. The computed




metrics indicate that, in general, the optimization approach (EO-$C_{m,2}$) improves bias, RMSE, and correlation, except for a
slight increase in bias for Gállego. Conversely, EO-$C_{m,1}$ produces poorer results, often yielding higher bias and RMSE than
L-$C_m$, except for Alpenrhein and Arve. As shown in Table C1, these trends generally persist across individual seasons, with a
few exceptions. Additionally, for the year not used in coefficient calibration (2022/23), EO-$C_{m,2}$ still demonstrates an overall
improvement in performance especially in reducing the bias and RMSE. However, the remaining errors stem from the fact that
the new coefficient enhances SCF during the melting season only, while errors persist during the accumulation season.

The poorer performance of EO-$C_{m,1}$ can be attributed to the fact that it is designed to perform well under conditions
where a pixel remains continuously snow-covered, with well-defined snow accumulation and melting periods. However, as
also stated by Pistocchi et al. (2017), this simplifying assumption may not hold in cases of intermittent snow cover or irregular
snow dynamics. The presence of intermittent snow might also be attributed to errors present in the satellite-based product,
particularly during the final melting phase when snow patches are present, leading to difficulties in applying the formula.

## 4.3 Effects on Water Balance

To fully assess the performance of the new melting coefficient in improving water balance estimates, spatialized SWE maps
would ideally serve as a reference. However, a comprehensive analysis is not feasible due to significant challenges in data
availability and usability. Given the spatial resolution of a LISFLOOD pixel, in-situ measurements are unreliable proxies,
as they fail to capture intra-pixel variability — particularly in basins with complex topography. Additionally, SWE data is
often missing, and when only snow height measurements are available, converting them to SWE requires assumptions about
snow density, introducing further uncertainty. Remote sensing products also have limitations. The accuracy of SWE derived
from microwave sensors is affected by factors such as vegetation cover, topography, and snow type (Pulliainen et al., 2020).
Furthermore, many available datasets, including those from the Copernicus Land Monitoring Service, have coarse spatial
resolutions and often exclude mountainous areas, further restricting their applicability (Takala et al., 2011). Therefore, we
propose an intercomparison with SWE estimates from two additional models in the Adige and in a subchatchment of the
Alpenrhein, along with an analysis based on monthly climatology in terms of LISFLOOD outputs for all the catchments.

Regarding the SWE intercomparison, we considered the IT-SNOW reanalysis dataset for the Adige basin (Avanzi et al.,
2024) (see Fig. 5). Additionally, for a small sub-basin of the Alpenrhein, the Dischma Valley, we compare our results with
the Swiss Operational Snow-Hydrological (OSHD) model system, available for the for the first five seasons (Mott, 2023; Mott
et al., 2023) (see Fig. 6). The metrics are reported in Table 3.

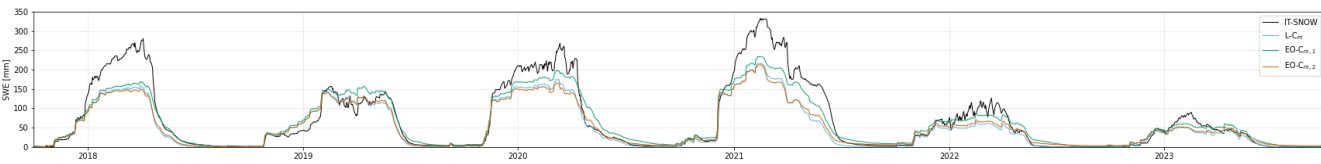

**Figure 5.** SWE time-series for the Adige river basin. In black the SWE from the IT-SNOW model, in cyan the SWE obtained with L-$C_m$, in
green the SWE obtained with EO-$C_{m,1}$ and in orange the SWE obtained with EO-$C_{m,2}$.




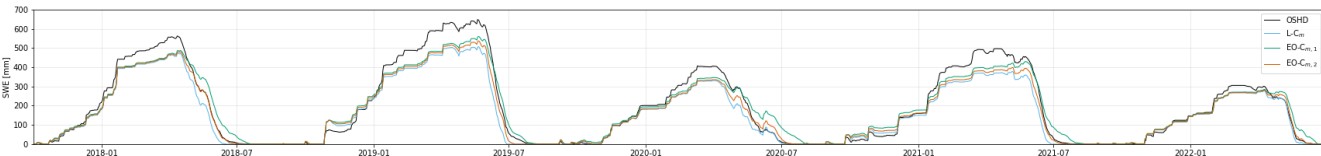

**Figure 6.** SWE time-series for the Dischma valley belonging to the Alpenrhein river basin. In black the SWE from the OSHD model, in cyan the SWE obtained with L-$C_m$, in green the SWE obtained with EO-$C_{m,1}$ and in orange the SWE obtained with EO-$C_{m,2}$.

**Table 3.** Evaluation of LISFLOOD SWE in terms of BIAS, RMSE, and correlation using IT-SNOW as the benchmark for Adige and OSHD for Dischma (Alpenrhein). The three different snowmelt coefficients L-$C_m$, EO-$C_{m,1}$ and EO-$C_{m,2}$ were utilized. Pixels where both the target and reference SCF are snow-free are excluded from the analysis.

|  | **L-$C_m$** | | | **EO-$C_{m,1}$** | | | **EO-$C_{m,2}$** | | |
|---|---|---|---|---|---|---|---|---|---|
|  | **Bias [mm]** | **RMSE [mm]** | **$\rho$ [-]** | **Bias [mm]** | **RMSE [mm]** | **$\rho$ [-]** | **Bias [mm]** | **RMSE [mm]** | **$\rho$ [-]** |
| Adige | $-53.93$ | 106.46 | 0.44 | $-55.24$ | 108.32 | 0.43 | $-35.32$ | 97.39 | 0.46 |
| Dischma | $-36.12$ | 76.66 | 0.96 | $-6.19$ | 79.62 | 0.94 | $-23.86$ | 75.11 | 0.96 |

The results confirm a generally good agreement but also highlight LISFLOOD's tendency to underestimate SWE compared to the other models in both basins. These differences are most likely linked to discrepancies in precipitation input data. Notably, IT-SNOW assimilates snow height measurements, which can enhance the accuracy of accumulation estimates. The SWE obtained with EO-$C_{m,2}$ falls between L-$C_m$ and EO-$C_{m,1}$. Especially for the Dischma Valley, the depletion curve is better

captured when using the new coefficient. Despite a lower bias obtained with EO-$C_{m,1}$ for Dischma, RMSE and correlation are improved with EO-$C_{m,2}$, which also appears to better represent the snow depletion by looking at the SWE trends. This aligns with expectations based on the SCA comparison. It should be noted that this analysis is not exhaustive, as it relies solely on intercomparison with other models and lacks validation against reference data.

To complete the assessment, we evaluated the changes in hydrological response resulting from the new coefficient. Since the

295 optimization method generally provided better results, we conducted this analysis using EO-$C_{m,2}$ only and compared it with benchmark simulations using the standard L-$C_m$. For this purpose, the LISFLOOD model was run from 1990 to 2022, with 2 warm-up years (1990-1991). The results are presented as monthly climatology between 1992 and 2022, derived from the monthly averages of the original 6-hourly model outputs 7. The outputs include SWE, snowmelt, total runoff, and discharge, all expressed in mm/month. The discharge climatology was calculated based solely on the dates with available observed data.

Dashed lines represent the 10th and 90th percentiles of the time series.





**Figure 7.** Monthly climatology of SWE, snowmelt, total runoff and discharge, in mm/month. In black, the climatology of observed river discharge, in blue the benchmark run calculated using L-$C_m$, and in orange the climatology from the run with the new calibrated coefficient EO-$C_m$.



In most of the catchments, the new snowmelt coefficient EO-$C_m$ influences the timing of snow accumulation and melting phases, which is, in some cases, reflected in the timing and magnitude of total runoff. In the Adige basin, the snow cover behavior is similar across both runs. However, the EO-run shows a slight shift in the timing of snowmelt, with its peak occurring in May, compared to the benchmark run where the peak is more evenly distributed between April and May. This shift has minimal impact on the generation of total runoff. Notably, the snowmelt coefficient assigned through the regionalization approach aligns closely with that derived from the EO-SCA. In the Alpenrhein, Arve, Gállego and Salzach basins, the new EO-runs show an increase in snow accumulation, driven by reduced snowmelt before the peak. This results in a higher magnitude of snowmelt and a shift in its timing, with the peak occurring later compared to the benchmark. The effect of the newly calibrated snowmelt coefficient extends to runoff generation, leading to increased runoff during the snowmelt phase and reduced runoff during the snow accumulation phase. In the Guadalfeo basin, a slight decrease in the snowmelt peak is observed, occurring in March, with no impact on total runoff. The model heavily underestimates river discharge compared to observations, the poor performance could be partially explained by the regionalization assignment of the parameters and/or by the fact that the Rules reservoir was opened in 2004, whereas is it has been included in the model for the whole simulation period (1992-2022). Therefore, the observed climatology considers 13 years in which the reservoir didn't exist, whereas the simulation considers that the reservoir was always there. In the Laborec basin, snow cover is reduced and snowmelt occurs earlier, peaking in February. Extreme values are reduced, with the 90th percentile decreasing from 60 to 52 mm for snow cover and from 100 to 80 mm/month for snowmelt. The total runoff increases in January and February but decreases in March. While the flow regime is broadly similar to observations, notable differences remain with observed runoff being 30% higher than both simulations. In the Swedish catchments, Mörrumsån and Umeälven, no significant differences in climatology are observed.

Performance statistics were computed using daily simulated and observed river discharge. Overall, the KGE values are similar between the two runs. However, slight improvements are observed for the Adige and Mörrumsån river basins, while KGE values for the Salzach, Laborec, and Alpenrhein river basins have decreased.





**Table 4.** Hydrological performance of current LISFLOOD configuration (benchmark) and LISFLOOD run with the new snowmelt coefficient

| | L-$C_m$ | | | | EO-$C_m$ | | | |
|---|---|---|---|---|---|---|---|---|
| | KGE [-] | Bias ratio [-] | $\rho$ [-] | Spread [-] | KGE [-] | Bias ratio [-] | $\rho$ [-] | Spread [-] |
| Adige | 0.75 | 0.96 | 0.76 | 1.04 | **0.76** | **0.97** | 0.76 | **1.03** |
| Alpenrhein | 0.73 | 0.75 | 0.90 | 1.01 | 0.70 | 0.75 | 0.89 | 1.12 |
| Arve | 0.80 | 0.83 | 0.89 | 0.99 | 0.80 | 0.83 | 0.89 | 0.99 |
| Gállego | −0.50 | 2.37 | 0.77 | 0.45 | −0.50 | 2.37 | 0.77 | 0.45 |
| Guadalfeo | −0.08 | 0.10 | 0.55 | 0.61 | −0.08 | 0.10 | 0.53 | **0.63** |
| Laborec | 0.74 | 0.98 | 0.75 | 0.93 | 0.69 | 0.98 | 0.70 | 0.93 |
| Mörrumsån | 0.76 | 0.87 | 0.81 | 0.98 | **0.77** | 0.87 | 0.81 | 0.98 |
| Salzach | 0.85 | 0.88 | 0.91 | 1.00 | 0.82 | 0.88 | 0.89 | 1.07 |
| Umeälven | 0.70 | 0.73 | 0.87 | 0.96 | 0.70 | 0.73 | **0.88** | **0.98** |

## 5 Discussion

In this study, we evaluated the LISFLOOD snow module and compared its current setup — which relies on traditional hydro-
logical calibration to fine-tune the snowmelt coefficient — with two alternative approaches that use dedicated calibration to
match satellite-derived snow cover data. Our findings reveal significant differences in the spatial distribution and magnitude of
$C_m$ across the different calibration approaches. Notably, LISFLOOD already produces reasonable SCA and SWE, particularly
when assessed at the basin level. However, calibrating the snowmelt coefficient separately leads to improvements, especially
in the representation of fractional snow cover and the depletion. Despite the substantial variations in the calibrated coefficients,
the LISFLOOD snow module demonstrates low sensitivity to these changes when evaluated in terms of climatology of the
water balance at catchment scale.

As expected, the optimization approach, which minimizes the error between the L-SCF and EO-SCF, yields more consistent
snow cover. It significantly improves the agreement between the L-SCF and EO-SCF, overcoming challenges posed by the
lack of continuous snow periods that might arise in the approach of Pistocchi et al. (2017). Notably, the optimized coefficient
improves the consistency of the melting phase with the depletion trends observed by the satellite, as shown in Fig. 4. This effect
is particularly evident in the Alpenrhein, Arve, and Salzach river basins, where a more linear snow season — characterized by
a single accumulation and melting phase — likely facilitates better alignment. Basins where ephemeral snow is more frequent
as Laborec and Mörrumsån show worse performances. However, it is important to discuss the reasons for further error sources.

A detailed analysis of the results indicates that these inconsistencies are primarily due to persistent cloud cover in the satellite
product, which can lead to reconstruction errors and inaccurate detection of snow cover. Additionally, forest coverage poses a
significant challenge for snow detection using optical data, as optical sensors are unable to detect snow beneath the canopy.





Discrepancies during the accumulation phase are likely due to inconsistencies in the input forcings, particularly in the estimation of solid precipitation. Note that the optimization process does not affect the accumulation phase, as SWE is independent of the melting coefficient for those days. Additionally, challenges in the SCF parameterization also contribute to these discrepancies, as accurately converting SWE to SCF remains a complex issue.

Furthermore, assuming a constant snowmelt coefficient over time may be an overly strong assumption. While seasonality is accounted for through the addition of $C_{seas}$, this might not fully capture variations in snowpack characteristics. The sinusoidal function that defines $C_{seas}$ merely adds a value in the range of $\pm 0.5\ mm/C° \cdot day$, without considering key geographical factors such as latitude, altitude, or aspect, which influence snowmelt. Factors such as differences in snow density, shallow snowpacks, or the presence of wind crusts, among others, can significantly affect snowmelt dynamics. This might also explain why, for the 2022/2023 season — excluded from the EO-based calibration — improvements are still present but are marginal compared to other seasons.

The evaluation in terms of hydrological variables leads us to conclude that no significant changes are observed. The new EO-based snowmelt coefficient impacts the timing of snow accumulation and melting phases, which, in some cases, is reflected in the timing and magnitude of total runoff. Although no significant differences are observed in the discharge climatology, the slight improvement/deterioration in KGE and its components suggest that daily discharge is affected, with a surprisingly slight improvement in a few catchments. Therefore, it is reasonable to assume that recalibrating the model with EO-$C_{m,2}$ as a non-calibrated parameter could lead to an improvement of the discharge simulation. The study confirms that traditional calibration provides a satisfactory average estimation of snowmelt dynamics at catchment level, which is beneficial for users aiming to understand the overall behavior of snow when only river discharge data is available. Although this method performs adequately, utilizing a per-pixel calibration offers the advantage of aligning the model more closely with specific snow cover observations. Isolating the contribution of snowmelt to river discharge would require a more detailed, event-based analysis, which is both challenging and beyond the scope of this study. Nonetheless, changes in the timing of peak accumulation and melting can have significant downstream effects, particularly in the most upstream regions of mountainous catchments with reservoirs, influencing both storage capacity and management strategies (Förster et al., 2016).

The method has been tested over a limited time frame and a small number of basins. Scaling its implementation to a continental scale presents several challenges, primarily due to the processing of a vast amount of data. The reference EO-SCA dataset was generated by combining HR and LR snow cover products, offering enhanced spatial detail while also increasing the complexity of managing multiple datasets. A large-scale implementation might require alternative ready-to-use datasets, such as those explored in Appendix A. Another possible approach could involve deriving the snowmelt coefficient based on current findings, leveraging pixel-wise features such as meteorology, geography, and land cover. However, in this study, no strong correlations were identified with topographical, geographical, or land cover features, as shown in Table C2 in Appendix C. The most influential feature appears to be elevation, which exhibits a negative correlation in the Adige, Arve, Salzach, and Guadalfeo river basins. This indicates that lower values of $C_m$ are estimated for higher elevations, meaning snow persists longer at higher elevations, as expected. For the Laborec and Umeälven basins, which are relatively flat, the correlation with elevation becomes positive. The correlation with mean snow cover duration also aligns with the findings for elevation, confirming that



snow lasts longer at higher elevations. However, a more in-depth investigation is needed to fully understand the underlying relationships and how the various features are inter-connected.

Based on our findings, we suggest proposing data assimilation schemes that not only modify the snowmelt coefficient, which impacts melting, but also account for local errors in the accumulation phase. Several assimilation schemes have been proposed that use snow cover information to improve hydrological models (Largeron et al., 2020). However, such an exercise requires the availability of a high-quality, daily gap-filled SCA time series. In other words, integrating EO products with hydrological models to leverage the strengths of each while addressing their limitations will be critical for improving the consistency and accuracy of snow modeling.

## 6 Conclusions

In this work, we assessed the performance and limitations of the LISFLOOD snow module in simulating snowmelt and snow water balances across diverse European watersheds. By leveraging a daily HR remote sensing snow cover product, our study provided valuable insights into the model's ability to reproduce accurate snow water balances and discharge predictions across varying climatic and topographical conditions.

We introduced the use of EO data, specifically derived from multi-source optical sensors. The daily gap-filled SCA, obtained from MODIS and Sentinel-2 data, was used to calibrate a spatially distributed snowmelt coefficient. We implemented and evaluated an appropriate snow cover parameterization for converting SWE into SCF to effectively evaluate snow depletion. Two methods were proposed for deriving the snowmelt coefficient: a previous approach, based on the number of snow-covered days (Pistocchi et al., 2017), and a novel optimization-based approach that minimizes the error between simulated and observed SCF.

Our main findings indicate that the accuracy of the current snow module, which has traditionally been calibrated using a classical hydrological approach, already produces satisfactory results when evaluated at basin scale. When compared against EO-SCF, the model's bias ranged from -0.56% for the Guadalfeo basin to 22.50% for the Laborec basin, with RMSE values varying from 20.43% for the Umeälven basin to 54.64% for the Mörrumsån basin. These results highlight the misrepresentation of fractional snow cover by the current LISFLOOD setup, which can be partially corrected for the melting days by applying a dedicated calibration of the degree-day. We obtained improvements up to 8% both in bias and RMSE when an optimization approach is used. This method outperforms the previous approach by Pistocchi et al. (2017), which yielded even poorer results w.r.t. the standard LISFLOOD setup. Despite this improvement, modifying the snow module while keeping prior values of the calibration parameters did not lead to substantial changes in performance when comparing discharge. The main differences were observed in the timing and magnitude of snow accumulation and melting, which sometimes affected the timing and magnitude of total runoff.

Shifts in snowmelt timing could have significant implications for water resources and storage. Ultimately, our findings highlight the potential of integrating models with satellite-based products to address existing limitations. By using more relevant observation data to refine specific calibration aspects, rather than relying solely on discharge, we can improve hydrological



modeling and water balance predictions. While this approach reduces the need for extensive hydrological model calibration, it still requires a complex calibration process involving large EO datasets. Further research is needed to develop a regionalized approach for the snowmelt coefficient. Nonetheless, in a changing climate where earlier snowmelt patterns are becoming more prevalent, integrating directly observed SCA could improve the prediction of evolving snowpack characteristics. This approach may provide a more robust framework for understanding and adapting to shifting snowmelt dynamics. Ultimately, the relevance

of the snow module performances is clearly dependent on the aim of the study and on the importance of the snow component in the water balance of the area.

*Code and data availability.* The source code of LISFLOOD model is available on GitHub at https://github.com/ec-jrc/lisflood-code. LISFLOOD parameters are available at https://jeodpp.jrc.ec.europa.eu/ftp/jrc-opendata/CEMS-EFAS/LISFLOOD_static_and_parameter_maps_for_EFAS/, whereas the meteorological forcings are available at https://jeodpp.jrc.ec.europa.eu/ftp/jrc-opendata/CEMS-EFAS/mete

orological_forcings/. The snow cover fraction time-series are available at https://zenodo.org/records/14961639. The code to reproduce the key results of this manuscript is available at https://github.com/vpremier/SCA4LISFLOOD.

## Appendix A: Earth Observation Snow Cover Dataset

In the methodological Section 2.1, we saw that the approach presented by Premier et al. (2021) to creating a continuous, gap-filled daily SCA dataset relies on two primary data sources, i.e., i) a LR SCF product based on optical data, and ii) HR optical

data. A widely used operational LR snow cover product with a long record (more than 20 years) is the MOD10A1 product derived from MODIS. Similarly, to obtain HR snow cover maps from Sentinel-2, we can utilize existing operational products such as the Copernicus Fractional Snow Cover (FSC) product. However, the use of other alternative datasets is also possible. Among others, we investigated the datasets listed in Table A1.

SnowFLAKES is also derived from Sentinel-2 and represents an alternative to FSC. It employs a more advanced algorithm,

specifically the Snow extent from applying a Flexible Learning Algorithm using KErnel-based Spectral unmixing developed internally at Eurac (Barella et al., 2022). The method is based on an unsupervised machine learning algorithm. While we have observed an improvement in snow detection, particularly in challenging situations such as shadowed pixels, the algorithm is still experimental, and the maps are not yet available for operational use. Similarly, an alternative to MOD10A1 but with shorter temporal coverage (from 2012 onward) is represented by VNP10A1, while ESC-H provides an alternative with coarser spatial

resolution. There are also operational products, such as GFSC and VNP10A1F, which provide gap-filled time-series to address cloud obstruction.

While many possible combinations are possible, we propose obtaining a daily gap-filled SCA by merging FSC and MOD10A1, referred to here as Copernicus derived gap-filled SCA (C-GSCA). This ensures both operational use and the possibility to generate data back in time. However, for some selected basins, we tested the combination of MOD10A1 with SnowFLAKES,

referred to here as SnowFLAKES derived gap-filled SCA (S-GSCA).





**Table A1.** Overview of snow cover products, their sources, native and resampled spatial resolutions, and temporal resolutions.

| Product | Description | Reference | Spatial Resolution | | Temporal Resolution |
|---|---|---|---|---|---|
| | | | Native | Resampled | |
| FSC | Fractional Snow Cover | Gascoin et al. (2019); Copernicus Land Monitoring Service (2021) | 20m | 2" | 5d |
| SnowFLAKES | Snow extent from applying a Flexible Learning Algorithm | Barella et al. (2022) | 20m | 2" | 5d |
| MOD10A1 | MODIS/Terra Snow Cover Daily L3 Global, V61 | Hall and Riggs (2021) | 500m | 20" | 1d |
| VNP10A1 | VIIRS/NPP Snow Cover Daily L3 Global, V2 | Riggs and Hall (2023) | 375m | 12" | 1d |
| ESC-H | Effective snow cover by VIS/IR radiometry | H-SAF Team (2020) | 0.01° | 1' | 1d |
| GFSC | Gap-filled Fractional Snow Cover | Copernicus Land Monitoring Service (2021) | 60m | 2" | 1d |
| VNP10A1F | Daily cloud-gap-filled VIIRS/NPP CGF Snow Cover L3 | Riggs and Hall (2022) | 375m | 12" | 1d |

Considering C-GSCA as the benchmark, we conducted an intercomparison exercise to assess the usability of different products, focusing on gaps primarily caused by cloud cover or satellite design. This analysis evaluates the potential of the analyzed datasets for use either as standalone inputs or within a merging approach by providing an overview of their agreement. Prior to the analysis, all products were resampled and aligned to a common grid to ensure comparability. Specifically, each
product was resampled to a resolution close to that of the original data and structured as an exact submultiple of the LISFLOOD grid, as shown in Table A1.

In Fig. A1, we provide an overview of the available data for the different basins, including the average percentage of cloud-covered pixels and the total number of available images for each season. Images that are completely obstructed by clouds over the study area are excluded from these metrics. Note that, for limiting the amount of downloaded data, not all the products are
available for all the basins or seasons. This does not change the outcomes of the analysis.





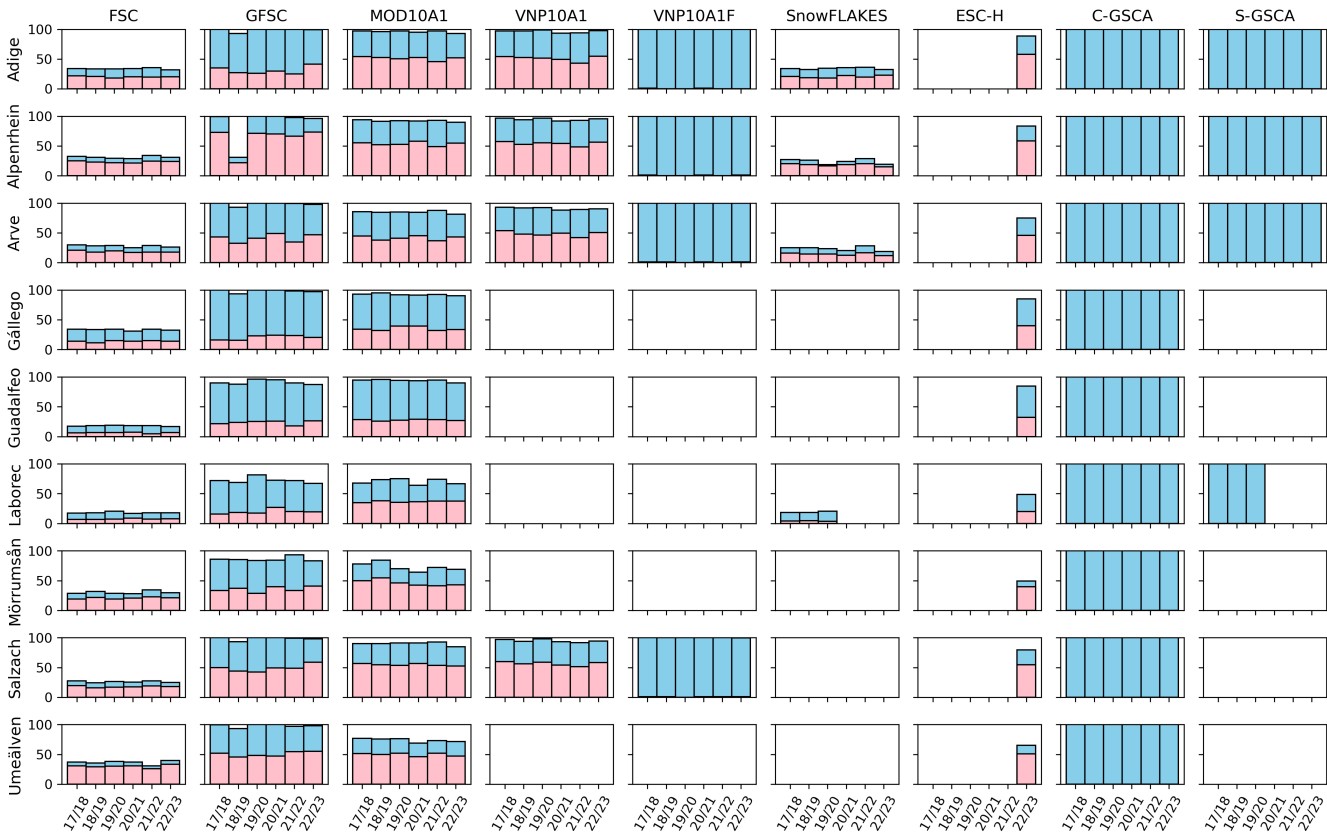

**Figure A1.** Overview of the available data for each basin and hydrological season. The percentage of available images throughout the year is shown in sky blue, while the percentage of cloud-covered pixels is represented in pink. Fully cloud-obstructed images are excluded from the calculations. Note that not all products were considered for every basin and season.





To assess the agreement or disagreement among the various datasets, we conducted an intercomparison exercise, calculating basic metrics such as bias, RMSE, and Pearson correlation coefficient ($\rho$) between pairs of datasets, using C-GSCA as the reference product. The results are reported in Fig. A2. For each matching date, metrics are calculated for SCF pixels aggregated to the final resolution of LISFLOOD. When the percentage of "no data" pixels exceeds approximately 10% within a LISFLOOD cell, SCF is marked as no data (equivalent to 90 pixels for FSC, GFSC, and SnowFLAKES; 3 pixels for VNP10A1 and VNP10A1F; and 1 pixel for MOD10A1). The figure shows the average metrics for the entire available period.





**Figure A2.** Boxplots of the resulting metrics (bias, RMSE and correlation) for each basin and product, using C-GSCA as the reference.



The results show that LR products (MOD10A1 and VNP10A1) offer similar image counts and metrics (on average, MOD10A1: bias 1.41%, RMSE 9.11%, and $\rho$ 0.73, VNP10A1: bias 0.27%, RMSE 14.50%, and $\rho$ 0.63). Note that MOD10A1 is used as input in the fusion algorithm to generate C-GSCA, but its values may be adjusted based on HR data. This results in a low bias but in the presence of outliers when compared to the benchmark (see Fig. A2). Given the similar results, VNP10A1 can be considered as a valid alternative to MOD10A1. ESC-H also exhibits comparable cloud obstruction, but shows the poorest agreement (worst metrics with bias -5.98%, RMSE 19.88%, and $\rho$ 0.47). The results suggest that its coarser spatial resolution reduces its utility as alternative product. In general, all datasets might not be used without a proper gap-filling algorithm.

The use of HR products (such as FSC and SnowFLAKES) is significantly constrained by their low revisit frequency, which leads to limited image availability over the year. Both products show similar levels of cloud coverage and image counts. Note that SnowFLAKES has fewer images due to the application of stricter criteria in cloud masking. The lowest availability of valid pixels for FSC is observed in the Umeälven basin, with only about 6% of pixels available throughout the year. In addition to cloud obstruction, this basin is also affected by polar night, leading to missing acquisitions from November through February. However, this limitation does not impact the algorithm's performance, as the basin remains fully snow-covered during this period, and acquisitions resume in time for the snowmelt season. Alpenrhein and Salzach follow, with approximately 8% of pixels available. In contrast, the basin with the highest availability of valid pixels is Gállego with a percentage of about 19%. Since C-GSCA is derived from FSC, we expect a good agreement (bias 0.09%, RMSE 0.38% and $\rho$ 0.99). Similar performance is also observed for SnowFLAKES (bias 0.64%, RMSE 7.11% and $\rho$ 0.87) and S-GSCA (bias 2.53%, RMSE 9.33% and $\rho$ 0.90), with differences due to the application of a different algorithm for the snow classification. In detail, the Adige basin exhibits a slightly higher bias. A detailed visual inspection revealed that, as also stated before, FSC tends to miss snow detection in shadowed areas. This is especially highlighted for basins with complex topography.

Regarding the gap-filled products, GFSC does not offer a valid alternative; although it provides a greater number of scenes, it still suffers from substantial "no data" gaps. The lowest availability is observed for Alpenrhein, with only around 25% of valid pixels. In fact, the gap-filling process primarily relies on propagating information from Sentinel-2 and Sentinel-1 data by using a defined temporal window (HR-S&I consortium, 2020). This is also the reason of similar performances w.r.t. the raw FSC (bias 1.18%, RMSE 10.17%, and $\rho$ 0.72).

In contrast, the nearly complete gap-filled data in the VNP10A1F product represents an attractive option with potential applications. A tendency toward underestimation relative to the reference is shown but the metrics remain reliable (bias -3.82%, RMSE 16.66%, and $\rho$ 0.78).

To conclude, we can state that C-GSCA is a product that by relying on HR data is expected to be more accurate w.r.t. LR data. Based on our findings, we decided to proceed with S-GSCA for the Adige basin while retaining C-GSCA for the remaining basins. In the main text, we generally refer to the time series as EO-SCA.



## Appendix B: Snow Cover Parametrization

As explained in Sec. 2.4, before comparing LISFLOOD results with the EO-SCA or computing the new snowmelt coefficient, an appropriate parametrization is required to convert SWE into SCF. Here we report the results obtained when considering the parametrization proposed by Swenson and Lawrence (2012) (see Sec. 2.3) compared with the approach proposed by Zaitchik and Rodell (2009). The following relationship is used:

$$
\text{SCF} = \min\left\{ 1 - \left[ \exp\left( -\tau \frac{\text{SWE}}{\text{SWE}_{\text{max}}} \right) - \frac{\text{SWE}}{\text{SWE}_{\text{max}}} \exp(-\tau) \right], 1 \right\} \tag{B1}
$$

where $\tau$ is the snow distribution shape parameter that relates the total amount of SWE to the SCF within the pixel. We set the snow distribution shape parameter $\tau$ to 4 globally, while $\text{SCF}_{\text{max}}$ varies from 13 mm for bare soil to 40 mm for forests as suggested by Zaitchik and Rodell (2009).

By taking the satellite-derived SCF as a benchmark, we evaluated the mean bias, RMSE, and correlation for each basin by comparing the EO-SCF with the SCF generated by LISFLOOD considering the standard coefficient L-$C_m$ under the two parametrization methods. The metrics are calculated pixel-wise and an average over time and space is computed. The results, detailed in Table B1, show improved metrics when adopting the parametrization proposed by Swenson and Lawrence (2012) for Adige, Guadalfeo and Umeälven while an improvement especially in terms of bias with the formula of Zaitchik and Rodell (2009) is shown for Alpenrhein, Arve, Laborec and Salzach while the other metrics do not show important differences. Given that the results show comparable performances and the fact that the parametrization by Swenson and Lawrence (2012) is more sophisticated, we decided to keep this approach. Therefore, all analyses and results in the main text are based on this parametrization.





**Table B1.** Evaluation of LISFLOOD SCF in terms of BIAS, RMSE, and correlation using the EO-SCF as the benchmark. The standard L-$C_m$ was utilized, with SWE converted to SCF using the parametrizations proposed by Swenson and Lawrence (2012) and Zaitchik and Rodell (2009).

|  | Swenson and Lawrence (2012) | | | Zaitchik and Rodell (2009) | | |
|---|---|---|---|---|---|---|
|  | **Bias [%]** | **RMSE [%]** | $\rho$ **[-]** | **Bias [%]** | **RMSE [%]** | $\rho$ **[-]** |
| Adige | 10.00 | 35.82 | 0.60 | 19.50 | 41.75 | 0.53 |
| Alpenrhein | −12.59 | 32.61 | 0.69 | −6.33 | 32.24 | 0.66 |
| Arve | −11.61 | 34.42 | 0.63 | −4.91 | 33.64 | 0.61 |
| Gállego | −0.56 | 44.46 | 0.52 | 2.74 | 42.03 | 0.57 |
| Guadalfeo | 7.66 | 28.46 | 0.32 | 14.26 | 33.21 | 0.33 |
| Laborec | 22.50 | 48.88 | 0.45 | 15.54 | 45.14 | 0.48 |
| Mörrumsån | 6.97 | 54.64 | 0.27 | −8.61 | 49.47 | 0.36 |
| Salzach | −6.83 | 36.84 | 0.67 | −2.17 | 34.50 | 0.68 |
| Umeälven | −1.06 | 20.43 | 0.58 | −2.07 | 21.77 | 0.58 |

## Appendix C:  Details on Snow Cover Area Evaluation

In this Appendix, we present further details that complement what presented in Sec. 4.2.

In Table C1, the metrics obtained for each hydrological season are reported. Again, the satellite-derived SCF is used as a benchmark. The metrics are calculated for three different cases: i) by using the standard LISFLOOD snowmelt coefficient L-$C_m$, ii) by considering the snowmelt coefficient obtained with the method proposed by Pistocchi et al. (2017) EO-$C_{m,1}$, and iii) by the snowmelt coefficient EO-$C_{m,2}$ obtained by applying an optimization. It should be noted that data from the 2022/23 season were not utilized in the derivation of the new degree-day coefficients.





**Table C1.** Evaluation of LISFLOOD SCF in terms of BIAS, RMSE, and correlation using the EO-SCF as the benchmark. The standard L-$C_m$ was utilized, with SWE converted to SCF using the parametrizations proposed by Swenson and Lawrence (2012) and Zaitchik and Rodell (2009).

| Basin | H.y. | L-$C_m$ | | | EO-$C_{m,1}$ | | | EO-$C_{m,2}$ | | |
|---|---|---|---|---|---|---|---|---|---|---|
| | | Bias [%] | RMSE [%] | $\rho$ [-] | Bias | RMSE [%] | $\rho$ [-] | Bias [%] | RMSE [%] | $\rho$ [-] |
| Adige | 17/18 | 11.19 | 34.28 | 0.61 | 17.93 | 32.78 | 0.64 | 9.61 | 31.18 | 0.62 |
| | 18/19 | 12.24 | 35.10 | 0.55 | 19.46 | 34.61 | 0.58 | 11.33 | 32.25 | 0.56 |
| | 19/20 | 8.04 | 31.58 | 0.59 | 16.36 | 31.65 | 0.59 | 5.53 | 28.56 | 0.60 |
| | 20/21 | 8.29 | 31.66 | 0.72 | 16.88 | 31.67 | 0.71 | 6.11 | 28.26 | 0.72 |
| | 21/22 | 6.85 | 33.18 | 0.59 | 16.36 | 31.80 | 0.60 | 7.20 | 30.54 | 0.61 |
| | 22/23 | 7.88 | 34.81 | 0.61 | 15.97 | 33.84 | 0.61 | 7.15 | 31.81 | 0.62 |
| Alpenrhein | 17/18 | −13.53 | 29.88 | 0.72 | −0.28 | 20.43 | 0.76 | −6.01 | 20.55 | 0.79 |
| | 18/19 | −12.50 | 30.91 | 0.71 | 1.67 | 22.39 | 0.76 | −4.94 | 22.49 | 0.78 |
| | 19/20 | −13.65 | 32.50 | 0.63 | 0.93 | 25.03 | 0.67 | −4.56 | 23.94 | 0.69 |
| | 20/21 | −12.73 | 32.46 | 0.69 | 3.48 | 25.56 | 0.71 | −4.03 | 23.96 | 0.74 |
| | 21/22 | −8.33 | 30.37 | 0.70 | 6.82 | 25.86 | 0.73 | 0.16 | 23.32 | 0.75 |
| | 22/23 | −15.73 | 35.36 | 0.61 | −2.48 | 25.39 | 0.70 | −7.91 | 26.88 | 0.69 |
| Arve | 17/18 | −8.56 | 30.73 | 0.64 | 8.07 | 28.97 | 0.65 | −2.63 | 26.11 | 0.65 |
| | 18/19 | −13.45 | 33.98 | 0.63 | 1.95 | 28.62 | 0.68 | −8.37 | 28.33 | 0.67 |
| | 19/20 | −10.43 | 29.87 | 0.54 | 3.33 | 25.22 | 0.59 | −4.11 | 24.03 | 0.57 |
| | 20/21 | −13.30 | 34.03 | 0.62 | 4.13 | 29.90 | 0.66 | −6.88 | 29.79 | 0.64 |
| | 21/22 | −8.61 | 35.40 | 0.61 | 10.86 | 32.40 | 0.67 | −1.25 | 28.81 | 0.66 |
| | 22/23 | −13.12 | 33.42 | 0.68 | 1.47 | 31.45 | 0.65 | −8.69 | 29.63 | 0.68 |
| Gállego | 17/18 | 26.89 | 44.14 | 0.36 | 28.37 | 42.74 | 0.38 | 23.86 | 41.83 | 0.37 |
| | 18/19 | 14.18 | 33.39 | 0.50 | 16.91 | 32.61 | 0.52 | 9.47 | 31.33 | 0.48 |
| | 19/20 | 21.21 | 29.41 | 0.46 | 25.69 | 30.49 | 0.45 | 22.48 | 29.31 | 0.41 |
| | 20/21 | −6.02 | 47.28 | 0.49 | −4.03 | 47.30 | 0.55 | −7.41 | 43.51 | 0.52 |
| | 21/22 | −2.27 | 27.49 | 0.56 | 0.91 | 26.98 | 0.56 | −6.36 | 27.02 | 0.55 |
| | 22/23 | 19.68 | 37.91 | 0.59 | 22.73 | 37.46 | 0.60 | 15.51 | 33.47 | 0.64 |
| Guadalfeo | 17/18 | 7.32 | 27.22 | 0.52 | 17.71 | 28.34 | 0.49 | 5.28 | 25.39 | 0.49 |
| | 18/19 | −3.81 | 32.46 | 0.30 | 6.00 | 33.02 | 0.26 | −4.77 | 32.10 | 0.32 |
| | 19/20 | −4.80 | 35.21 | 0.37 | 16.48 | 34.29 | 0.48 | −7.16 | 34.14 | 0.34 |
| | 20/21 | 5.35 | 32.19 | 0.47 | 21.28 | 34.61 | 0.44 | 3.56 | 30.58 | 0.46 |
| | 21/22 | 3.15 | 32.69 | 0.32 | 16.95 | 31.67 | 0.44 | 2.37 | 31.05 | 0.31 |
| | 22/23 | 22.87 | 33.06 | 0.27 | 30.80 | 37.62 | 0.30 | 23.08 | 32.90 | 0.25 |
| Laborec | 17/18 | 10.74 | 45.61 | 0.41 | 14.27 | 46.00 | 0.41 | 4.32 | 45.98 | 0.40 |
| | 18/19 | 12.42 | 36.54 | 0.68 | 16.99 | 38.69 | 0.66 | 1.95 | 34.60 | 0.68 |
| | 19/20 | 34.98 | 57.97 | 0.15 | 35.37 | 58.17 | 0.14 | 32.90 | 57.52 | 0.15 |
| | 20/21 | 12.78 | 42.55 | 0.56 | 16.59 | 45.73 | 0.51 | 4.48 | 39.70 | 0.60 |
| | 21/22 | 30.38 | 48.46 | 0.44 | 34.55 | 51.12 | 0.41 | 19.81 | 42.95 | 0.49 |
| | 22/23 | 45.04 | 61.43 | 0.42 | 46.96 | 62.38 | 0.42 | 39.21 | 58.30 | 0.43 |
| Mörrumsån | 17/18 | 5.40 | 44.79 | 0.35 | 8.85 | 45.69 | 0.34 | 4.56 | 44.62 | 0.35 |
| | 18/19 | 35.33 | 53.22 | 0.51 | 40.67 | 57.10 | 0.49 | 34.22 | 52.21 | 0.53 |
| | 19/20 | −10.06 | 71.13 | -0.29 | −3.62 | 71.01 | -0.31 | −11.98 | 70.68 | -0.28 |
| | 20/21 | −13.14 | 44.62 | 0.52 | −6.70 | 39.58 | 0.58 | −14.91 | 46.22 | 0.49 |
| | 21/22 | 22.59 | 56.23 | 0.25 | 28.39 | 57.94 | 0.26 | 21.68 | 55.62 | 0.26 |
| | 22/23 | 6.24 | 62.84 | -0.01 | 12.52 | 62.21 | 0.01 | 3.18 | 63.16 | -0.02 |
| Salzach | 17/18 | −1.92 | 36.28 | 0.69 | 10.48 | 29.18 | 0.69 | 4.79 | 27.35 | 0.73 |
| | 18/19 | −8.01 | 30.71 | 0.75 | 4.56 | 26.14 | 0.71 | −3.32 | 25.56 | 0.73 |
| | 19/20 | −6.15 | 37.76 | 0.66 | 10.42 | 31.94 | 0.69 | 3.66 | 30.24 | 0.71 |
| | 20/21 | −14.44 | 38.90 | 0.62 | 0.08 | 30.38 | 0.66 | −7.24 | 31.45 | 0.66 |
| | 21/22 | −2.35 | 34.51 | 0.67 | 12.34 | 30.42 | 0.70 | 5.16 | 26.43 | 0.73 |
| | 22/23 | −6.43 | 40.40 | 0.63 | 9.41 | 31.58 | 0.69 | 1.77 | 31.56 | 0.69 |
| Umeälven | 17/18 | −0.02 | 19.98 | 0.59 | 2.08 | 18.23 | 0.65 | 0.56 | 18.84 | 0.62 |
| | 18/19 | −0.82 | 20.05 | 0.65 | 2.34 | 17.13 | 0.72 | 0.10 | 17.71 | 0.70 |
| | 19/20 | −1.40 | 16.31 | 0.75 | 0.27 | 12.70 | 0.82 | −1.08 | 13.77 | 0.79 |
| | 20/21 | −2.78 | 16.38 | 0.65 | 0.31 | 13.89 | 0.68 | −1.13 | 13.40 | 0.71 |
| | 21/22 | 0.28 | 17.06 | 0.56 | 2.71 | 14.63 | 0.65 | 0.97 | 13.34 | 0.65 |
| | 22/23 | −3.61 | 26.00 | 0.48 | 0.34 | 21.85 | 0.55 | −2.08 | 22.57 | 0.54 |

Yearly scatterplots for each basin are shown in Fig. C1 and C2.





**Figure C1.** Scatterplots of the L-SCA obtained with the L-$C_m$ (in cyan), EO-$C_{m,1}$ (in green) and $C_{m,2}$ (in orange) against EO-SCA for the Adige, Alpenrhein, Arve, Gállego, and Guadalfeo.




**Figure C2.** Scatterplots of the L-SCA obtained with the L-$C_m$ (in cyan), EO-$C_{m,1}$ (in green) and EO-$C_{m,2}$ (in orange) against EO-SCA for the Laborec, Mörrumsån, Salzach and Umeälven.

In Table C2, the Pearson correlation coefficients between EO-$C_{m,2}$ and topographical and meteorological features, such as elevation (DEM), standard deviation of the elevation (DEM $\sigma$), slope, forest coverage, mean snow cover duration (SCD), mean precipitation (P) and temperature (T) - where the average is calculated for the six analysed hydrological seasons - are reported.





**Table C2.** Pearson correlation coefficients between the EO-$C_{m,2}$ and topographical, land cover and meteorological features.

|  | DEM | DEM $\sigma$ | Slope | Forest | SCD | P | T |
|---|---|---|---|---|---|---|---|
| Adige | −0.58 | −0.22 | −0.05 | 0.52 | −0.75 | 0.13 | 0.55 |
| Alpenrhein | −0.28 | −0.41 | −0.16 | −0.06 | −0.36 | 0.35 | 0.18 |
| Arve | −0.50 | −0.48 | −0.32 | 0.06 | −0.54 | −0.42 | 0.47 |
| Gállego | −0.04 | −0.01 | −0.01 | 0.40 | −0.27 | −0.03 | 0.00 |
| Guadalfeo | −0.77 | −0.15 | −0.06 | 0.65 | −0.70 | −0.36 | 0.72 |
| Laborec | 0.45 | 0.16 | 0.25 | 0.39 | 0.32 | 0.40 | −0.48 |
| Mörrumsån | −0.03 | 0.10 | 0.08 | 0.21 | −0.06 | −0.02 | −0.05 |
| Salzach | −0.55 | −0.68 | −0.39 | −0.06 | −0.69 | −0.42 | 0.50 |
| Umeälven | 0.45 | 0.00 | 0.10 | −0.57 | 0.37 | 0.68 | −0.37 |



*Author contributions.* FM and AP designed and conceptualized the study together with VP and CM. VP wrote the paper based on input and feedback from all coauthors. VP conducted the experiments and processing related to the SCA and the estimation of snowmelt coefficients. FM performed the experiments and processing associated with the LISFLOOD simulations. All authors contributed to the research design,

as well as the discussion, analysis, and interpretation of the results.

*Competing interests.* The authors declare that they have no conflict of interest.

*Acknowledgements.* This research was supported by funding from the Joint Research Centre (JRC) through Tender No. JRC/IPR/2023/VLVP/2678 - "Support to LISFLOOD model development: testing of the snow module". We would like to acknowledge the work of our colleague Riccardo Barella (Institute for Earth Observation, Eurac Research) for providing the SnowFLAKES product, whose development was supported

by the ESA Snow CCI project (Contract No. 4000124098/18/I-NB) and the ESA EXPRO+ AlpSnow - Alps Regional Initiative project (Contract No. 4000132770/20/I-NB), both financed by the European Space Agency (ESA). We also thank the CEMS Hydrological Data Collection Centre for providing the historical river discharge data.



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
