# Peer review of "Assessing the Impact of Earth Observation Data-Driven Calibration of the Melting Coefficient on the LISFLOOD Snow Module"

_EGUsphere, 2025_

## Author Comment (AC1)

Review paper egusphere-2025-2157

**Assessing the Impact of Earth Observation Data-Driven Calibration of the Melting Coefficient on the LISFLOOD Snow Module**

By Premier et al.

We thank the Anonymous Reviewer for providing valuable feedback. We believe that the manuscript can be improved by considering his suggestions and by clarifying several critical aspects that were not previously well explained. Below, we provide our point-by-point responses, highlighted in red.

**General Comments**

This study investigates the calibration of the snowmelt coefficient in the LISFLOOD hydrological model using Earth Observation (EO)-derived snow cover data. The authors propose two EO-based calibration methods and assess their impact on snow cover fraction (SCF), snow water equivalent (SWE), and discharge simulations across nine European river basins. The manuscript contributes to ongoing efforts to integrate remotely sensed data into large-scale hydrological modeling.

We thank the Reviewer for recognizing the topic importance.

However, several methodological ambiguities, design inconsistencies, and literature gaps limit the manuscript's clarity, reproducibility, and broader relevance. The introduction focuses heavily on LISFLOOD while neglecting to situate the work within the substantial body of existing literature on EO-based snow calibration and assimilation techniques, many of which have long addressed multi-objective calibration using snow and streamflow data. The authors should better articulate the novelty of their approach beyond its application to LISFLOOD.

We thank the Reviewer for the thoughtful comments and for highlighting areas where the manuscript could benefit from improved clarity and contextualization. We acknowledge that the introduction currently focuses predominantly on LISFLOOD, and we agree that it could be expanded in a revised version of this manuscript. However, as explained in the following responses, the emphasis on LISFLOOD reflects the practical scope of our study—since it is the model that is relevant within our operational context.

We believe that the main novelty of this work lies in assessing the effects of a more consistent representation of a specific LISFLOOD component—the snow module—on hydrological response. To this end, we calibrated the snowmelt coefficient using EO data and re-ran the model using the EO-derived snowmelt coefficient, with all other settings consistent with the EFAS setup. However, the same approach can be applied to any other hydrological model.

This approach allowed us to directly evaluate the behavior of the snow module. Our analysis shows that adjusting the snowmelt coefficient to better represent the snow cover did not necessarily require recalibrating the other parameters. While the original model was already accurate for snow cover when considering the entire basin, this adjustment provided a more accurate agreement at the pixel scale when evaluating SCF. Crucially, these improvements in SCF did not alter the resulting streamflow.

We acknowledge that we did not develop an entirely new methodology, as many of the techniques used are established in the literature. However, we combined these techniques in a novel way to evaluate the LISFLOOD snow module differently from previous studies (e.g., Thirel et al., 2012; Pistocchi et al., 2017). Our approach includes a novel calibration technique made possible by the use of a newly developed, gap-filled high-resolution snow cover time series, which is expected to have higher accuracy than commonly used gap-filled SCA products. This is largely due to the integration of high-resolution data, which we elaborate on later in the answers.

[Figure]

*Figure 1 Workflow overview.*

In this graphical abstract, we present our approach. We derive detailed daily SCF information from satellite data, which serves as a benchmark in our workflow. Concurrently, the LISFLOOD model is run, and the resulting SWE is converted to SCF using an appropriate parameterization. In this process, the snowmelt coefficient (Cm) is treated as a free parameter and calibrated through a pixel-wise optimization on a yearly basis. Finally, a pixel-wise average is computed. One season from the dataset is excluded from the optimization and reserved solely for evaluation purposes.

While this results in a post-replacement of the snowmelt coefficient, it could be possibly integrated upstream and result in a better representation of a specific module of the model.

We would also like to highlight that we initially considered submitting this work as a technical note. However, a HESS associated editor pointed out in such a case, the format would have required

significant shortening and potentially omitted key methodological details. We chose the full research article format to provide a more comprehensive presentation of our methods and findings.

Should we be given the opportunity to revise the manuscript, we are confident that we can improve its clarity, reproducibility, and broader relevance by addressing these points.

This is particularly important given that the improvements achieved with EO-calibrated snowmelt coefficients remain modest, or even questionable, with respect to discharge simulations. This raises broader concerns about the hydrological value and operational significance of the proposed methodology.

We thank the Reviewer for raising this important point. Indeed, we recognize that the improvements in discharge simulations achieved through EO-calibrated snowmelt coefficients are modest and, in some cases, may appear limited. However, we can obtain a more accurate representation of the snow cover fraction (SCF).  This said, we try to address a broader methodological question: *should multi-objective calibration (e.g., streamflow + SCA) be pursued, or are alternative strategies that aim at a more realistic representation of SCA feasible*?

Our findings suggest that a sequential calibration strategy—where the snowmelt coefficient is first adjusted upstream using EO-derived Snow Cover Area (SCA), followed by downstream calibration on streamflow—can serve as a viable and potentially more efficient alternative to full multi-objective calibration. In addition, we demonstrate that a standard calibration on streamflow, followed by a targeted post-adjustment of the snowmelt coefficient based on SCA, can still yield acceptable performance without the need for recalibrating the full model.

Moreover, several critical aspects of the methodology—such as the SWE–SCF parameterization, spatial resolution strategy, calibration procedure, and test basin selection—are poorly explained, inconsistently justified, or insufficiently analyzed. The comparison between models, methods, and performance metrics is often difficult to follow and underinterpreted. Key information for reproducibility (e.g., calibration configurations, data preprocessing protocols) is also lacking.

Many of the methodological components applied in this study (e.g., SWE–SCF parameterization, calibration procedures) are based on previously published and validated approaches. For the sake of brevity and to avoid unnecessary repetition, we opted to refer to those sources and keep the descriptions concise. In a revised version, we will aim to strike a better balance between brevity and completeness.

While the topic is relevant and the integration of EO data into hydrological modeling remains important, the manuscript in its current form suffers from fundamental methodological opacity, weak novelty positioning, and limited hydrological impact. Key sections are unclear or

poorly justified, the experimental design is inconsistent, and the results do not support the claimed contributions. For these reasons, I recommend rejection. A revised version would require a substantial restructuring of both the methodology and the scientific framing to meet the standards expected for publication in HESS.

We thank the Reviewer for highlighting the weaknesses of the manuscript. We are confident that by addressing these points in the revised version, we can significantly improve the quality and clarity of the work.

**Specific Comments**

**L14–15**: This sentence is too vague to provide meaningful insight into the methodology or key contributions.

We thank the Reviewer for this comment. We agree that the original sentence "These findings highlight the potential of integrating EO data to enhance snowmelt simulations and improve water balance predictions, with important implications for hydrological modeling and water resource management" is too vague. The sentence will be changed as "These findings highlight the potential of integrating EO data to calibrate the snow melt coefficient without changing other calibration parameters. This approach may offer practical advantages in situations requiring accurate snow cover representation, although our results also show that standard calibration procedures provided in this case an acceptable representation of snow dynamics".

**L15–75**: The introduction overfocuses on LISFLOOD and insufficiently addresses broader research on EO-based calibration and snow data assimilation. The authors should frame the novelty of their study in light of widely used multi-objective calibration approaches and explain how their work differs in terms of technique or purpose—not merely model specificity.

We thank the Reviewer for this comment. However, we believe that an explanation of LISFLOOD in the introduction is necessary, as it is the model we use and central to our analysis. The emphasis on LISFLOOD reflects the practical scope of our study—it is the tool available and relevant for our operational context.

That said, the core objective of this work is to address a broader methodological question: *should multi-objective calibration (e.g., streamflow + SCA) be pursued, or are alternative strategies feasible*?

Our findings suggest that a sequential calibration approach—first adjusting the snowmelt coefficient upstream using EO-derived SCA, followed by downstream calibration on streamflow—can be a viable and potentially more efficient alternative. Furthermore, we find that a standard calibration on streamflow alone, followed by a post replacement of the snowmelt coefficient based on SCA, can still yield acceptable results without the need to recalibrate the entire model.

Although our experiments were conducted using LISFLOOD, we believe the rationale and calibration strategy explored here are applicable to other similar distributed hydrological models.

In response to this comment, we will revise the introduction to better highlight the broader methodological implications of our work and clarify how our approach differs from conventional multi-objective calibration strategies.

**L59–60**: The use of snow data in hydrological model calibration is not new and has become a common practice for over a decade. The statement should be revised to reflect this context.

Thanks for the comment. We will rephrase the sentence from "This calibration approach differs from traditional hydrological calibration methods by introducing an independent process that does not rely solely on discharge data" to "This study investigates the effects of adjusting the snowmelt coefficient with a post replacement of the snowmelt coefficient based on EO snow cover data without the need for a complete recalibration of the entire model." also based on the previous answer.

**L66–67**: The role of Sentinel-2 data here is unclear. If its main function is downscaling MODIS, this should be stated explicitly. Furthermore, cloud-free MODIS products (e.g., MOD10A1 Version 6) and well-documented gap-filling techniques are already available—please clarify why these were not used or compared.

We thank the Reviewer for the insightful comment, which allows us to clarify the role of high-resolution data in our methodology. While a simplified interpretation might suggest that Sentinel-2 data are used solely to downscale MODIS, this is not entirely accurate. As described in Premier et al. (2021), Sentinel-2 data are also employed to correct MODIS observations. Our method relies on the assumption that high-resolution (HR) data are more accurate than low-resolution (LR) data, particularly for fractional snow cover, where Sentinel-2 can detect snow patches that MODIS may miss.

Unlike other state-of-the-art approaches that only downscale MODIS while preserving its Snow Cover Fraction (SCF), our method also corrects SCF, offering a novel improvement. While we acknowledge the existence of various gap-filling and downscaling algorithms in literature, we

consider the validation of our methodology beyond the scope of this paper, as it has already been published in Premier et al. (2021).

That said, we agree with the Reviewer on the importance of comparing our method with other existing and operational products. For this reason, Appendix A includes a comparison with the gap-filled VNP10A1F product, which demonstrates that it can serve as a good alternative to our more complex and labor-intensive algorithm. Although MOD10A1F could also be tested, we expect similar performance due to its comparable characteristics and gap-filling strategy. Additionally, VNP10A1F offers a slightly higher spatial resolution, which is advantageous.

**L110–115**: Calibrating 14 parameters without detailed justification seems excessive. A summary table of parameters and ranges is essential. A sensitivity analysis would help assess the importance of the snowmelt coefficient relative to other parameters and reveal possible interdependencies.

The table below specifies the 14 model parameters typically fitted in a LISFLOOD calibration. The model has more parameters (for instance, the temperature thresholds that define precipitation as snowfall or the start of snowmelt), but this is the selection of the most sensitive parameters after years of working with the model. Not all parameters are sensitive in all the catchments, e.g., the reservoir parameters are not in a catchment without reservoirs, or the snowmelt coefficient in areas where snowfall never occurs. These cases are identified by the calibration tool and the irrelevant model parameters are removed from the calibration. Further details can be found here: https://ec-jrc.github.io/lisflood-code/4_annex_parameters/.

Table 1. Calibration model parameters in the OS-LISFLOOD model.

| Process | Parameter | Range Minimum | Default | Maximum | Unit |
|---|---|---|---|---|---|
| Snow | Snowmelt coefficient | 2.5 | 4 | 6.5 | mm/°C·day |
| Soil | Xinanjiang b | 0.01 | 0.5 | 5 | - |
| | Preferential flow | 0.5 | 4 | 8 | - |
| | Groundwater percolation | 0.01 | 0.8 | 2 | mm/day |
| | Upper GW zone constant | 0.01 | 10 | 40 | days |
| | Lower GW zone constant | 40 | 100 | 1000 | days |
| | Lower GW zone threshold | 0 | 10 | 30 | mm |
| | Groundwater loss | 0 | 0 | 0.5 | mm/day |
| Streamflow routing | Manning's n modifier: main channel | 0.5 | 1 | 2 | - |

| | | | | | |
|---|---|---|---|---|---|
| | Mainning's n modifier: floodplain | 0.5 | 1 | 2 | - |
| | Trigger of split routing | 0 | 2 | 20 | - |
| Reservoirs and lakes | Reservoir: normal storage | 0.01 | 0.8 | 0.99 | - |
| | Reservoir: normal release | 0.25 | 1 | 2 | - |
| | Lake multiplier | 0.5 | 1 | 2 | - |

Regarding the purpose of this research, the independent calibration of the snowmelt coefficient does not interfere with the rest of the model parameters. On the contrary, that is what sequential calibration tries to avoid. In the "traditional" calibration, the snowmelt coefficient would be calibrated together with the other 13 parameters to streamflow. In that scenario, due to equifinality, it may happen that the calibrated snowmelt coefficient does not correctly reproduce snow processes, but the streamflow simulation performs well. That is actually what we wanted to explore with this study.

**L111**: Please clarify whether model calibration is based solely on streamflow using KGE. If so, this should be justified in light of the study's stated focus on snow processes.

This paragraph explains the usual LISFLOOD calibration, where the snowmelt coefficient is fitted together with all the other parameters to maximize the Kling-Gupta Efficiency of the streamflow in a gauging station downstream. It is introduced here for reference, as a comparison with the sequential approach explored in the analysis. We will make clearer that this paragraph refers to the "traditional" calibration.

**L123–124**: If all 14 parameters are optimized per sub-basin against streamflow, why isolate the snow module for analysis? The risk of equifinality and parameter interactions should be acknowledged and discussed.

We thank the Reviewer for the comment and would like to clarify the calibration routine of LISFLOOD. The L-Cm snowmelt coefficient was calibrated as part of the LISFLOOD calibration for the European domain within the European Flood Awareness System (EFAS) and the European Drought Observatory (EDO). This calibration routine involves 14 parameters, and is based solely on streamflow using KGE, as detailed on the CEMS page:
https://confluence.ecmwf.int/display/CEMS/EFAS+v5.0+-+Calibration+Methodology+and+Data
In this study, LISFLOOD itself was not recalibrated. The evaluation of the module structure and the calibration routine were not within the scope of our work.

Given that the model calibration is based exclusively on streamflow and involves multiple parameters, it can lead to satisfactory streamflow reproduction but does not guarantee an accurate representation of snowmelt processes. This limitation may arise due to parameter interdependencies, model complexity, equifinality, and the common challenge in distributed hydrological modeling of "being right for the wrong reasons" (Beven and Cloke, 2012).

For this reason, we decided to calibrate the snowmelt coefficient using EO data and then re-run the model with only the EO-snowmelt coefficient different from the EFAS setup.

This process allowed us to evaluate how the snow module behaved.

What we found out is that by changing the Snowmelt coefficient, we observe a quasi-equifinal for streamflow but not as equifinal for SCA, SWE and melting. The catchment average runoff and the discharge at the outlet have low sensitivity to the changes in the snowmelt coefficient, effects of the new parameters in upper basins are expected (as we will show later), but the snowmelt coefficient calculated using the traditional calibration compensates for local differences. If we work with a model where this holds true, we can calibrate the snowmelt coefficient to our will and then calibrate the rest independently, thus arguably achieving higher accuracy on snow but very similar on streamflow. We show that for LISFLOOD this holds true.

We acknowledge that sensitivity analyses have been conducted for LISFLOOD:

parameter uncertainty ( https://www.tandfonline.com/doi/pdf/10.1623/hysj.53.2.293 )

global sensitivity analysis (https://www.gdr-mascotnum.fr/media/mascot13zambrano-poster.pdf)

calibrated parameter analysis:

https://www.sciencedirect.com/science/article/pii/S0022169418307467?via%3Dihub

sensitivity analysis

(https://www.sciencedirect.com/science/article/pii/S2214581816300817?via%3Dihub)

global sensitivity and uncertainty analysis

(https://www.sciencedirect.com/science/article/pii/S0022169417301671)

While we agree with the Reviewer that a detailed sensitivity analysis would be valuable, especially in the context of LISFLOOD, we consider it beyond the scope of the current study, which focuses primarily on the snow module and the calibration of the snowmelt coefficient.

We will add a subsection in the methodology section where we address the valuable comment of the reviewer:

**2.5 Evaluation against current LISFLOOD setup**

To complete the assessment, we evaluated the changes in hydrological response resulting from the snowmelt coefficient (Cm) that performed best in simulating the SCA. Specifically, we replaced only the snowmelt coefficient originally estimated during the EFAS calibration (ECMWF, 2022) (L-Cm), with the one calibrated using Earth Observation (EO) data (EO-Cm), and ran the LISFLOOD model from 1990 to 2022, including two warm-up years (1990–1991). We stressed that the model was not re-calibrated for the other 13 parameters, allowing us to isolate the impact of the Cm on river discharge in the selected catchments and to assess whether the current EFAS setup can be

trusted to realistically model snow accumulation and melting dynamics. The snowmelt coefficient in LISFLOOD is traditionally calibrated as part of a multi-parameter routine that optimizes 14 parameters simultaneously against observed streamflow data, focusing on maximizing the Kling-Gupta Efficiency (KGE). However, this approach is prone to equifinality, where multiple parameter combinations yield similar discharge performance (Beven, 2006), but may mask inaccuracies in the representation of other processes because of parameters interactions and errors compensation. Consequently, while EFAS calibration achieves good streamflow fits, it can produce less realistic simulations of processes not directly constrained by streamflow data (Beven, 2019).

This issue is especially pronounced in large-scale hydrological models applied at continental and global scales, where structural uncertainty becomes significant due to the application of a uniform model framework across catchments with widely varying climatic and physical characteristics (Beven, 2006; Beven and Cloke, 2012).

By calibrating only the Cm with EO data independently from the full 14-parameter calibration, we directly constrain the snowmelt process using spatially and temporally explicit snow observations, reducing the equifinality problem and compensatory effects among parameters. Replacing only the Cm in the LISFLOOD model while maintaining the remaining parameters fixed allows for the evaluation of two critical aspects: (1) the extent to which the full multi-parameter calibration accurately reproduces snow accumulation and melt dynamics, and so if the model is fit-for-purpose for snow related evaluations. And (2) the influence of the independently calibrated Cm on river discharge, specifically assessing its impact on model performance as measured by the Kling-Gupta Efficiency (KGE).

The results are presented as monthly averages between 1992 and 2022, derived from the mean monthly values of the original 6-hourly model outputs. The outputs include snow water equivalent (SWE), snowmelt, total runoff, and discharge, all expressed in mm/month. The discharge monthly average was calculated based solely on dates with available observed data. Dashed lines in the figure represent the 10th and 90th percentiles of the time series, providing insight into the variability and uncertainty of the modeled hydrological response."

**L137**: The description of elevation banding is unclear. How are elevation classes defined and implemented at the 1.4 km model resolution, which significantly smooths real terrain features? What are the implications for snow accumulation and melt representation?

We thank the Reviewer for the comment. We agree that our explanation of how elevation zones were defined was not sufficiently clear. Due to the relatively coarse resolution of LISFLOOD cells (1', ~1.4 km), significant sub-pixel variability in snow accumulation and melt can occur, particularly in areas with large elevation differences within a single pixel.

To address this, snow processes are modeled separately within three elevation zones defined at the sub-pixel level. These zones are determined based on a normal distribution of elevation values, which has been shown to represent well the actual distribution. To this purpose, the standard deviation of elevation within a grid cell is calculated from the Multi-Error-Removed Improved-Terrain (MERIT) DEM with a spatial resolution of 90 m. The three elevation zones—A, B, and C—are each assumed to cover one-third of the pixel area.

Assuming that the average pixel temperature corresponds to the mean pixel elevation, temperatures for the lower zone A and upper zone C zones are estimated by applying a fixed lapse rate (L = 0.0065 °C/m) to the elevation differences from the mean. Snow accumulation and melt are then modeled separately for each zone, using the temperature at each zone's centroid as a proxy for local conditions.

To improve clarity, we will add these details that can be found in the LISFLOOD model official documentation (https://ec-jrc.github.io/lisflood-model/2_04_stdLISFLOOD_snowmelt/) to the manuscript.

**L140–164**: The SWE–SCF parameterization is central but confusing. Equations (7), (8), and (10) appear circular or contradictory. Their derivation, purpose, and assumptions must be clarified. Also, kaccum plays a key role but is not explained. A graphical illustration would help. The brief mention of the Swenson & Lawrence vs. Zaitchik & Rodell methods lacks depth and justification.

We thank the Reviewer for this comment, which lets us clarify the rationale and implementation of the SWE-SCF parametrization approach. First, regarding our choice of parameterizations, we are aware that several approaches have been proposed in literature. A non-exhaustive list that can be included in the manuscript is:

- Luce, C. H., Tarboton, D. G., & Cooley, K. R. (1999). Sub-grid parameterization of snow distribution for an energy and mass balance snow cover model. Hydrological Processes, 13(12-13), 1921-1933.
- Douville, H., Royer, J. F., & Mahfouf, J. F. (1995). A new snow parameterization for the Météo-France climate model: Part I: validation in stand-alone experiments. Climate Dynamics, 12(1), 21-35.
- Roesch, A., Wild, M., Gilgen, H., & Ohmura, A. (2001). A new snow cover fraction parametrization for the ECHAM4 GCM. Climate Dynamics, 17, 933-946.
- Liston, G. E. (2004). Representing subgrid snow cover heterogeneities in regional and global models. Journal of climate, 17(6), 1381-1397.
- Niu, G. Y., & Yang, Z. L. (2007). An observation-based formulation of snow cover fraction and its evaluation over large North American river basins. Journal of geophysical research: Atmospheres, 112(D21).

- Helbig, N., van Herwijnen, A., Magnusson, J., & Jonas, T. (2015). Fractional snow-covered area parameterization over complex topography. Hydrology and Earth System Sciences, 19(3), 1339-1351.
- Pimentel, R., Herrero, J., & Polo, M. J. (2017). Subgrid parameterization of snow distribution at a Mediterranean site using terrestrial photography. Hydrology and Earth System Sciences, 21(2), 805-820.

While we acknowledge the importance of a comprehensive treatment of SCF parameterizations, this lies beyond the main scope of our study, which is to propose an alternative calibration approach for the snowmelt coefficient in the LISFLOOD model. To that end, we chose to test two parameterizations that offer a balance between model complexity and data availability. Swenson & Lawrence is also an approach widely used in the Community Land Model (CLM). On the other hand, Zaitchik & Rodell is a simpler empirical method requiring fewer input data, yet shown to produce consistent results. Our results (Table B1) show that both parameterizations yield similar performance in our experimental setup, with a general better agreement with the EO data when using Swenson & Lawrence.

Regarding the equations, they are derived from the mentioned publication and from the code of the CLM model, available here:
CTSM/src/biogeophys/SnowCoverFractionSwensonLawrence2012Mod.F90 at master · ESCOMP/CTSM

While we agree that we could better explain the meaning and role of the equations, we consider a full derivation of the SWE–SCF parameterization beyond the scope of this paper, as it would add considerable complexity and potentially distract from the main objectives. However, we provide here a brief conceptual explanation to help clarify the approach.

The accumulation formulation (Eq. 7) is based on the probability that a pixel becomes snow-covered after a precipitation event. Specifically, the snow-covered fraction (s) is defined as:

$$s = \min(1, \, k_{accum} \cdot SWE)$$

This defines *s* as the probability that a pixel is snow-covered, with *kaccum* acting as a scaling parameter that relates SWE to fractional coverage. Accordingly, the probability that a pixel remains snow-free is **p = 1 − s**.

If multiple snowfall events occur, and assuming independence (i.e., uncorrelated events), the cumulative probability that a pixel remains snow-free is the product of the individual **p** values. Therefore, after *N+1* events, the snow-covered fraction can be updated as:

$$SCF_{N+1} \; = \; 1 \; - \; (p_{N+1})(p_N) \; = \; 1 - (1 - s_{N+1})(1 - s_N)$$

Similarly, Eq. 7 that is also implemented in CTSM/src/biogeophys/SnowCoverFractionSwensonLawrence2012Mod.F90 at master · ESCOMP/CTSM, is based on a probabilistic interpretation involving a *tanh* function, where *tanh* ensures that SCF asymptotically approaches 1 as SWE increases.

Regarding the depletion curve (melting), Equation 8 is derived empirically, as stated by the original authors. It is important to note that Equation 10 can be obtained by inverting Equation 8. Additionally, in the original paper, Equation 11 is reported with a typographical error; however, the correct formulation is implemented in the corresponding code. We have also been in contact with the original authors to confirm that we are using the correct version of the formula. Please, for deeper understanding check:

Swenson, S. C., & Lawrence, D. M. (2012). A new fractional snow-covered area parameterization for the Community Land Model and its effect on the surface energy balance. Journal of geophysical research: Atmospheres, 117(D21).

We agree that, as stated in the original paper, the accumulation parameter kaccum plays an important role. For this reason, we chose not to keep it fixed. Instead, we calculate it dynamically using our EO-derived SCF data at the time of the first snow accumulation, as also suggested by the original authors. This parameter represents the ratio between SCF and SWE at the onset of accumulation—when the pixel is still only partially snow-covered—and is therefore essential for determining the rate or "speed" of snow accumulation.

**L165**: Is the snowmelt factor calibrated independently of other LISFLOOD parameters? If so, a discussion of the implications and potential benefits of multi-objective calibration (including SCA and runoff) is needed.

We thank the reviewer for this insightful comment. Indeed, the snowmelt coefficient was calibrated independently from the other LISFLOOD parameters, and no multi-objective calibration was performed in this study. The primary objective of comparing the EFAS setup with the same setup incorporating the independently calibrated snowmelt coefficient was to evaluate whether recalibrating this single parameter would significantly impact the hydrological cycle. Additionally, this approach enabled us to assess the robustness of the full LISFLOOD calibration—which involves 14 parameters—in accurately reproducing snow dynamics.
We acknowledge that this aspect was not clearly articulated in the manuscript. To address this, we will consider to include a dedicated paragraph in the Methodology section discussing the implications of independent versus multi-objective calibration strategies, highlighting the potential benefits of jointly calibrating snow cover area (SCA) and runoff. Furthermore, we will expand the

Discussion section to incorporate these considerations and their relevance to model performance and parameter interactions.

**L169**: The snow balance equation is invalid in glaciated basins where annual melt can exceed snow accumulation due to negative mass balances. The method should either exclude these basins or account for ice dynamics.

We thank the Reviewer for this valuable comment. The basins that include glaciers are: Adige, Alpenrhein, Arve, and Salzach. The basins without glacierized areas are: Gallego, Guadalfeo, Laborec, Mörrumsån, and Umeälven.

Although the glacier-covered area is relatively small in most of the glacierized basins—less than 1% of the total area (approximately 0.9% for Adige and Salzach, and 0.6% for Alpenrhein)—we acknowledge that glaciers can still have a non-negligible influence, particularly in the Arve basin, where the glacierized area is approximately 5%. This influence is also visible in our results (see Figure 4 and discussion starting at Line 247).

We agree that the ice component has not been adequately addressed in the current version of the manuscript. Our initial intention was to mask out pixels where the glacier coverage exceeded a certain threshold during the calibration of the snowmelt coefficient. A proper representation of glaciated areas would require distinguishing between snow and ice surfaces and applying different coefficients accordingly. However, we believe this is beyond the scope of our work.

Furthermore, the LISFLOOD setup does not explicitly model glacier dynamics. The standard approach in LISFLOOD adjusts melt rates for ice-covered areas using a sinusoidal function to increase melt in summer, accounting for enhanced radiation and changes in surface albedo. However, this is a simplified treatment that does not capture true ice mass balance or dynamics.

**L173–176**: The intent of this paragraph is unclear. Please rephrase to clarify what is being estimated or illustrated.

Thank you for your comment. The purpose of the paragraph is to clarify the conditions under which Eq. 11 is valid. Specifically, this equation assumes a single, continuous snow period—defined as a sequence of days during which a pixel remains continuously snow-covered. In such cases, it is reasonable to assume that total accumulation equals total melt over the snow season, ignoring other processes like wind or gravitational snow transport.

However, in some pixels—especially at lower elevations or in temperate climates—multiple snow periods may occur (e.g., snow melts and re-accumulates later). In these cases, Eq. 11 should ideally be applied separately to each distinct snow period. While this would be more accurate, it would also introduce additional methodological complexity.

For consistency with the original approach proposed by Pistocchi et al., (2017), we retain their simplification of applying the equation across all snow-covered days, regardless of whether snow cover is continuous or intermittent. This simplification is a known limitation of the method and one reason we expect improved performance from the optimization-based approach proposed in this study.

We propose rephrasing the paragraph as follows to make this point clearer: "Eq. 11 is strictly valid only over a single, uninterrupted snow period—defined as a sequence of days when the pixel remains continuously snow-covered—we follow the approach of Pistocchi et al. (2017) and apply the equation across all snow-covered days, regardless of continuity. This simplification avoids additional complexity that would arise from segmenting and analyzing multiple snow periods per pixel. However, we acknowledge this as a limitation of the method, particularly in lower-altitude regions where snow accumulation and melt cycles occur more frequently within a single season."

**L178**: What is being compared here? A model simulation using observed SCFs? The terminology and structure are confusing and require clarification.

We thank the Reviewer for this comment. By L-SCF (LISFLOOD SCF), we refer to the snow cover fraction (SCF) estimated using the LISFLOOD model. The LISFLOOD model itself does not directly provide SCF as an output; instead, it outputs snow water equivalent (SWE), which is computed using Equations 4 to 6, as a function of the snowmelt coefficient Cm.

To derive SCF from the modelled SWE, a parameterization (Equations 8–10) must be applied. This parameterization derives SCF from SWE to SCF, thus being SCF a function of Cm too.

The snowmelt coefficient Cm is treated as a free parameter in our framework and is subject to optimization. To optimize it, we minimize the error between L-SCF and EO-SCF, which refers to the SCF derived from Earth Observation (EO) data and serves as a reference.

**L165–191 (Section 2.4)**: This section should be rewritten to clearly explain both EO-based methods for estimating melt factors. The current text lacks transparency and methodological rigor.

Thanks for the comment. For the sake of clarity, we will revise the section and add information about parameter ranges, the combination of the different hydrological seasons, and how the algorithm works in snow-free pixels.
Also please refer to Figure 1 to have a general overview of our approach.

The resolution mismatch between EO data (50–500 m) and model grid (1.4 km) introduces major issues. Downscaling MODIS to 50 m and then reaggregating to 1.4 km is not clearly justified. How are orographic gradients in precipitation and temperature accounted for at

this coarse scale? The authors should better discuss whether a semi-distributed approach (e.g., elevation bands) or higher-resolution modeling would improve consistency with EO data and SWE estimates.

Thank you for the comment. As also mentioned in a previous response, our methodology is not straightforward downscaling of MODIS data. Instead, it includes a correction step aimed at addressing known limitations of the MODIS sensor, such as errors due to grain size variability, solar zenith angle, viewing geometry, and atmospheric effects that are not fully accounted for in the retrieval algorithm. Following the approach described in Premier et al. (2021), we do not treat the MODIS-derived snow cover fraction (SCF) as an absolute value. Instead, we interpret it within a "safety belt" of uncertainty and primarily rely on high-resolution data to reconstruct snow patterns through robust statistical analysis. These reconstructed patterns implicitly account for topographic effects, including orographic influences. While the snow cover retrieval method is not the main focus of this paper, we agree that this aspect deserves a clearer explanation and will clarify it in a revised version.

Regarding orographic gradients, as noted in a previous response, our model accounts for elevation-dependent snow processes by dividing each pixel into three elevation zones. This semi-distributed approach allows us to represent variations in snow accumulation and melt processes with altitude. Also note that meteorological forcing (EMO-1) considers the temperature gradient with altitude. That's not the case for precipitation.

We agree, however, that higher-resolution modeling could improve consistency between model outputs and EO-derived SWE, particularly in complex terrain. Nevertheless, it is important to stress the fact that the current model has been developed to run at continental scale (resolution > 1km). Increasing the resolution would increase computational time and output data size, crucial considerations for a model that runs operationally. Increasing resolution does not guarantee improved performance in all model compartments, since some processes, now simplified or ignored, might become more relevant and higher resolution (Van Jaarsveld et al., 2025). Moreover, we would like to emphasize that fine-resolution modeling is only as accurate as the quality of the input forcing data. In many basins—especially at high elevations—observational data are scarce or of limited accuracy, which poses challenges for high-resolution modeling. In contrast, EO observations may capture some processes, such as wind redistribution of snow, more effectively when high-resolution acquisitions are available. Thus, the integration of EO data remains a valuable complement to physically based modeling.

The manuscript suggests the key research question is spatial calibration (pixel vs. basin scale, L47–48), but this is insufficiently explored. How do calibration results differ at each spatial scale? What is gained or lost?

Thank you for this important comment. As shown in Figures 2 and 3, the pixel-wise calibration might result in a distribution that highly differs from that of the lumped coefficients. Initially, our idea was to investigate whether snowmelt coefficients calibrated at pixel scale could reveal meaningful patterns or correlations with topographic, geographic, or land cover features. Our initial guess was that such correlations might eventually allow for snowmelt parameterization based on spatial characteristics alone—potentially reducing reliance on EO data, which are often labor-intensive to process. However, as shown in Table C2 and discussed in Line 370, our analysis did not reveal strong correlations between the calibrated snowmelt coefficients and those spatial features. We are also aware that while pixel-level calibration allows for more spatial detail and potentially better alignment with EO-derived patterns, it increases computational burden. Basin-scale calibration has shown in this study to be already sufficiently robust but may hide local heterogeneity.

**L202–209 & Table 1**: The basin selection lacks justification. Several catchments (e.g., Arve, Salzach) include glaciers, while others (e.g., Guadalfeo) are subject to strong anthropogenic influences (e.g., reservoirs, diversions). These factors are not modeled and introduce significant uncertainty. Their inclusion must be explained and justified—or their results treated with caution.

We thank the Reviewer for this comment. As already stated in Section 3, the basins were selected based on the following criteria: i) they are all snow-dominated catchments, ii) they represent a range of geographical contexts, iii) they cover diverse land cover types, and iv) they span a range of elevation zones. Additionally, as mentioned in L204 of the manuscript, the initial selection also considered the availability of discharge data.

We acknowledge the Reviewer's point that some of the selected catchments, such as the Arve and Salzach (influenced by glacier melt), and Guadalfeo (subject to significant anthropogenic influence), include processes that introduce additional sources of uncertainty. In the specific case of the Guadalfeo River, the Rules reservoir—constructed in 2006, midway through our analysis period (1990–2020)—is included in EFAS. However, EFAS assumes the reservoir was present throughout the simulation period, which introduces bias in the model output for this catchment. Additionally, the Guadalfeo basin was not part of the EFAS calibration, which further contributes to the relatively poor model performance shown in Figure 7. These limitations will be clearly acknowledged in the revised discussion to help contextualize the results.
For a detailed explanation of the reservoir modeling approach in LISFLOOD, we refer the Reviewer to the official LISFLOOD documentation:
https://ec-jrc.github.io/lisflood-model/3_03_optLISFLOOD_reservoirs/
And for a comprehensive list of reservoirs included in EFAS, refer to:
https://hess.copernicus.org/articles/28/2991/2024/

Table 1 / Calibration vs. Regionalization: It is unclear why some basins are calibrated while others are regionalized. This methodological inconsistency needs to be explained. Consistent baseline comparisons are essential to interpret calibration effectiveness.

We thank the reviewer for the comment. Some basin parameters came from a regionalization approach and not calibration because of the lack of river discharge observation. The map of the domain calibrated with river discharge is presented here:
https://confluence.ecmwf.int/display/CEMS/EFAS+v5.0+-+Calibration+Methodology+and+Data

[Figure]

*Figure 2  In blue the area of the pan-European domain for which discharge observations were available; in yellow the area of the pa-European domain for which discharge observations were NOT available for EFAS v5 calibration. The area in black is not included in the modeling domain.*

The Adige and Guadalfeo basins were not calibrated during the calibration of the EFAS system; their parameters were assigned using the regionalization methodology from Beck et al. (2016) [L120].
For this study, we managed to get observed data of river discharge, so the comparison against the simulated river discharge from LISFLOOD was possible.
As much as this could be seen as inconsistent, we believe that this was actually an opportunity to evaluate the regionalization approach effectiveness, a common challenge in data-scarce/ungauged basins. We will stress this better in the methodology and discussion

**Figure 1**: The coarse DEM resolution leads to incorrect hypsometry (e.g., Arve basin's maximum elevation is ~4800 m a.s.l., not 3700 m). This smoothing likely affects snow (and ice) accumulation and melt modeling and should be discussed.

Thank you for the comment. We agree that the resolution is very coarse, but LISFLOOD has been developed as a large-scale model. However, as discussed previously in another answer, the intrapixel variability is partially considered by using three elevation zones inside the pixel. Assuming that the average pixel temperature corresponds to the mean pixel elevation, temperatures for the lower zone A and upper zone C zones are estimated by applying a fixed lapse rate (L = 0.0065 °C/m) to the elevation differences from the mean. Snow accumulation and melt are then modeled separately for each zone, using the temperature at each zone's centroid as a proxy for local conditions.

**L204–209**: The temporal alignment of model forcing (1992–2022) and snow data (2017–2023) is confusing. Are independent evaluation years used? If so, how is calibration/control separation ensured? A proper split-sample test would strengthen the study.

Thanks for the comment. The snowmelt coefficient has been calibrated over a five-season period, from October 1, 2017, to September 30, 2022. The sixth hydrological season, from October 1, 2022, to September 30, 2023, is used only for evaluation purposes.  This period was chosen being the period of maximum availability of Sentinel 2 data (as stated from L205). After using the previous EO seasons to calibrate the snowmelt coefficient, we ran LISFLOOD in the period 1992-2022 to compare the effects of the differently calibrated snowmelt coefficients in the SWE climatology. We might include also the last seasons to have more consistent periods, however we believe this is not affecting our outcomes.

**L216 and Throughout**: The manuscript uses many overlapping abbreviations for calibration methods (e.g., EO-Cm, EO-Cm1, EO-Cm2, LBFGS-B, L-Cm) with insufficient explanation. This confuses readers. Provide a summary table of methods and a glossary of acronyms. Terms should be redefined when introduced in different sections.

We thank the Reviewer for his comment. In a future version, we will add a summary table and redefine the terms in each section.

**L220–225**: The differences in results across basins should be discussed. Are certain physiographic features (e.g., elevation, land cover, glacier presence) associated with better or worse performance?

We thank the Reviewer for raising this interesting point, which inspired us to carry out an additional analysis. For the sake of brevity, we report here the results in terms of RMSE, evaluated for SCF derived from both L-Cm (Figure 3) and EO-Cm,2 (Figure 4). The performances are analyzed against selected physiographic features (mean elevation, forest coverage, and slope) and climatic features (mean precipitation, temperature, and snowfall). Furthermore, we distinguish basins with and without glaciers using different colors.

The results show a tendency for higher errors in lower-elevation and flatter catchments, while increased forest coverage is also associated with higher errors. Regarding the climatic features, an inverse relationship with RMSE is observed: basins with higher precipitation and snowfall tend to show lower errors. This is expected, since basins with less precipitation — especially less solid precipitation — have more ephemeral snow cover, leading to a higher fraction of partial snow cover and thus greater potential for errors.

Glacierized catchments do not appear to show substantial differences compared to non-glacierized ones. It is also noteworthy that the Umealven basin consistently stands out as an outlier with lower RMSE. This may be explained by its prolonged and complete snow cover, which likely results in more stable snow conditions and fewer errors.

We can include these additional results in a future version of the manuscript.

[Figure]

*Figure 3 Performances in terms of SCF RMSE versus different physiographic and climatic features when using L-Cm.*

[Figure]

*Figure 4  Performances in terms of SCF RMSE versus different physiographic and climatic features when using EO-Cm,2.*

**L270**: This section is mischaracterized as a "water balance" analysis, but it is actually a comparison of LISFLOOD SWE with other model outputs. The full hydrological balance (precipitation, evapotranspiration, storage changes) is not analyzed, which would be relevant.

We thank the reviewer for the comment. We agree that the terminology is misleading since we are not reviewing the water balance of the model.
We will create 2 separate subsections. The first one "4.3 Snow Water Equivalent Exploratory Evaluation" will cover the comparison against other models that estimate SWE, the other subsection will be called "4.4 effects on LISFLOOD long-term simulation" and it will cover the comparison of the performance of the LISFLOOD model run against the LISFLOOD results using the EO-Cm. The monthly (and daily will be added) averages for discharge, snowmelt, snow cover and total runoff will be presented. Moreover, we will include a spatial comparison of river discharges that will show us in which areas of the catchment the discharge is affected by the EO-Cm.
Regarding the analysis of the other components of the model, our analysis focused on the impact of the snowmelt coefficient on discharge. Moreover, given the fact that LISFLOOD is a mass balance model, the effects on other components are limited, especially looking at the limited impact of EO-Cm on total runoff, shift (decrease) in infiltration/evapotranspiration are expected in

Salzach, Arve, and Alpenrhein for the months of June-July given the higher runoff compared to the EFAS simulation.

The limited impact of the new snowmelt factor at catchment scale will be further analysed in the revised manuscript where we will include the following analysis that looks at the impact of the EO-Cm at sub-catchments level, and for upstream areas above 100 km2. The impact of EO-Cm is visible locally, and more precisely in some sub-catchments. This corroborates our thesis in saying that the current EFAS calibration serves its purpose in representing well catchment average snowmelt dynamics. However, users should be careful when using simulated discharge upstream of the calibrated stations, since river discharge, in some cases, can be very different between the discharge from EFAS5 and the discharge computed using EO-Cm.

Those differences are shown in the plots below, when we computed the Normalized Euclidian distance (NED) between the EFAS river discharge and the EO-snowmelt coefficient of river discharge. River discharge was selected for upstream areas above 100 km2 for both model outputs and min-max normalized. The NED was then computed between grid cells at the same location.

The NED is presented for each catchment in Figure 5. Darker colors represent grid cells/river sections where the two models produce similar daily discharge outputs (lower NED); lighter colors indicate areas where the models diverge more significantly.

The spatially heterogeneous EO-snowmelt coefficients have a noticeable local impact in some river reaches with low upstream area, here the differences between model outputs are more pronounced. However, this influence decreases progressively downstream as localized effects are smoothed out along the flow path. By the time the discharge reaches calibration points, typically located further downstream, the impact becomes negligible, as the calibration process compensates for or overrides local parameter variations.

This is confirmed also by looking at the daily and monthly averages at catchment level, where differences between the two discharges are negligible, besides Salzach, Arve, and Alpenrhein. Therefore, while users can trust the current EFAS5 version to represent catchment-scale snowmelt-runoff dynamics, we recommend caution when interpreting simulated river discharge in upstream or mountainous areas, especially where inflow to reservoirs is critical, as local snowmelt-runoff processes may not be fully captured. These points will be clarified and supported with references in the revised manuscript.

[Figure]

[Figure]

*Figure 5 normalized Euclidean distance between daily discharge from the EFAS5 model and the discharge from the LISFLOOD model run with the new EO-coefficient. In grey the catchment area. In magenta the station used in the evaluation in section.*

**L279–281**: Comparing LISFLOOD SWE with other models without harmonized forcing data is misleading. The comparison should be framed as qualitative or exploratory—not as validation.

We completely agree with the reviewer that the analysis does not represent a formal validation. As stated in L282 of the manuscript, we refer to the analysis as an intercomparison of existing SWE products. We acknowledge that this analysis is not exhaustive and should be considered a preliminary step, especially given the lack of reference SWE datasets (as stated in L292-293). Even if in-situ SWE measurements were available, they would not provide a suitable reference due to the coarse resolution of the LISFLOOD model and the high intra-pixel variability introduced by complex topography. Additionally, other models cannot be considered absolute references, as they may have inherent limitations stemming from model parameterizations and the quality of forcing inputs. We will clarify in the revised version of the manuscript that this is intended as an exploratory analysis rather than a comprehensive validation.

**Figure 7 and L319 etc.**: "Climatology" is misused. Use "seasonal average" or "mean monthly values." Also, explain what the envelopes in the figure represent. Monthly aggregation may obscure important daily dynamics—consider showing daily averages instead.

We agree with the reviewer, and we will amend the terminology in the manuscript. We will add daily averages as shown in the following plots.

[Figure]

[Figure]

[Figure]

**Metrics Reporting**: The interpretation of metrics (e.g., RMSE, KGE) lacks depth. What does a specific improvement mean in operational or hydrological terms? A summary table of relative improvements across basins would aid comparison.

We thank the reviewer for the comment. Given the limited differences between the 2 performances of the LISFLOOD model we do not believe that an extra table is necessary.
We will include in the result a better description of the metrics. Such as: Bias is practically identical, which means that water is not stored nor lost between the two runs. The correlation coefficient is slightly worse in some catchments, which means that the time of the peak flows is slightly hindered. The variability has mixed outcomes.
The decreased performance in correlation is the most significant in operational terms. A lower correlation coefficient suggests that the timing of peak flows is somewhat less accurately captured. This is particularly important in the operational context of EFAS5, where changes in correlation directly impact the system's ability to anticipate or delay peak flows, an aspect that is critical for effective flood hazard communication and early warning.
It should be stressed however that model outputs from the LISFLOOD run using the EO-snowmelt coefficient were not recalibrated. If the model were to be used operationally, it would undergo a calibration round where 13 parameters ought to be calibrated, the original ones (14) minus the snowmelt coefficient. After the re-calibration of the model, the new metrics should be analyzed in terms of their impact in operational terms.

**L295–300**: This methodological content appears out of place in the results section, indicating a need for clearer structure throughout.

We thank the Reviewer for the comment. We will move the paragraph to the Methodology section.

**L325–385**: The discussion should better engage with existing literature on snow data assimilation. It is widely recognized that improvements in snow state representation do not always lead to improved streamflow prediction. This should be acknowledged and contextualized.

We thank the reviewer for the comment; however, our study does not assimilate the SWE state into the model. The SCA is converted into SWE and this value is used to calibrate the snowmelt coefficient at pixel level, as further explained in the previous answers. We will clarify better in the revised version of the manuscript.

**L345 & L380**: The SWE–SCF conversion is treated superficially. Other formulations exist and should be discussed. Additionally, the distinction between calibration and data assimilation should be made clearer, especially if the authors position their method as a calibration approach.

Thank you for the comment. We have already provided additional details on the SWE–SCF parameterization in our previous response. In the revised manuscript, we will better justify the choice of the Swenson and Lawrence, (2012) formulation and mention alternative approaches available in the literature.

Regarding the distinction between calibration and data assimilation, we position our method as a calibration approach, since the snowmelt coefficient is statically optimized and kept constant across multiple hydrological years. That is, we use EO-SCF to calibrate a model parameter (the snowmelt coefficient), rather than dynamically adjusting the model state during runtime.

However, we also recognize that the methodology could be extended to a data assimilation framework, in which EO-SCF is assimilated in near real-time to update model states based on the observed EO-SCF.

**Model Structure**: The limited impact of improved melt factors on discharge suggests structural limitations in LISFLOOD (e.g., degree-day assumptions, decoupling of snow and runoff). These issues deserve more attention in the discussion.

We thank the reviewer for the comment.
The degree-day method is a very simple, conceptual approach that is broadly used in other large-scale hydrological models, such as PCRGlob, CWatM or mHM. Even though the model might benefit from an improved representation of the snow processes, we believed that, given the model scale and purpose, LISFLOOD is able to satisfactory capture the main processes. We will,

however, mention in the manuscript that more sofisticated models should be tested in future, also taking into condiration the higher resolution of this model version (1', ~1.4 km) compared to the previous one (5km).
Regarding the limited impact of the EO-cm on discharge, we believe that this is generally true when looking at the discharge at the station outlet (with some seasonal differences in Salzach, Arve, and Alpenrhein basins).  As shown in the Normalized Euclidean Distance plots, local differences in discharge are present upstream.

**L408–409**: This conclusion is not strongly supported by the preceding results and should be revised or qualified.

We thank the reviewer for the comment. We agree with the comment, and we will amend the conclusion accordingly. Our study highlights that calibrating with Snow Cover Area from EO can improve local dynamics, but that for the scale of the basin analyzed the impact on river discharge simulation is comparable with the parameters estimated by the traditional LISFLOOD calibration.

The manuscript would benefit from careful revision for clarity, structure, and language. Sections are often dense and overly technical, with insufficient explanation of key decisions. A clearer narrative structure, consistent terminology, and simplified figures would greatly improve readability.

We fully agree with the Reviewer that the manuscript can be further improved thanks to their constructive comments. We hope to have the opportunity to revise it accordingly, based on these responses.

---

## Author Comment (AC2)

Review paper egusphere-2025-2157 (Reviewer Francesco Avanzi)

**Assessing the Impact of Earth Observation Data-Driven Calibration of the Melting Coefficient on the LISFLOOD Snow Module**

By Premier et al.

We thank Dr. Francesco Avanzi for providing valuable feedback. We believe that the manuscript can be improved by considering his suggestions and by clarifying several critical aspects that were not previously well explained. Below, we provide our point-by-point responses, highlighted in red.

Premier and colleagues have presented an improved calibration approach for the LISFLOOD snow module that is based on leveraging high resolution satellite data. This approach minimizes the error between observed and simulated snow cover fraction, and directly impacts the simulation of SWE. Authors test this method across a variety of study catchments in Europe and provide a detailed analysis of the improvements in snow simulation, as well as an interesting water balance perspective.

This paper is technically sound, and the effort of using high resolution satellite data into a large scale hydrological model is relevant and interesting for HESS. At the same time, there remain some aspects that could be expanded and improved. I still see value in this manuscript, and thus I am recommending a major revision.

We thank the Reviewer for recognizing the  importance of the topic and for providing valuable feedbacks.

My main point is that using snow data in addition to streamflow data in calibrating a hydrologic model has been widely explored (see for example https://www.tandfonline.com/doi/abs/10.1080/01431161.2010.483493, https://www.sciencedirect.com/science/article/abs/pii/S0022169419312132?via%3Dihub, https://www.sciencedirect.com/science/article/abs/pii/S002216941300320X, https://hess.copernicus.org/articles/26/5627/2022/hess-26-5627-2022.html, https://www.sciencedirect.com/science/article/abs/pii/S0022169424013167). As a result of significant research in the hydrologic community over the last 15 years, a multi-objective calibration that involves at least snow and streamflow data is now considered state of the art. Meanwhile, even doing so does not necessarily imply an improvement in model performance.

We acknowledge that multi-objective calibration has been widely explored in the literature. In the revised manuscript, we will better position our work within the context of state-of-the-art approaches, including citations to the references provided by the reviewer in the Introduction section.

As clarified in our response to Anonymous Reviewer 1, our work addresses the methodological question: should multi-objective calibration (e.g., streamflow + SCA) be pursued, or are alternative strategies that aim for a more realistic representation of SCA feasible?

Our findings suggest that a sequential calibration strategy—where the snowmelt coefficient is first calibrated upstream using EO-derived SCA, followed by downstream calibration on streamflow—can offer a viable and potentially more efficient alternative to full multi-objective calibration. Furthermore, we show that a standard calibration based solely on streamflow, followed by a targeted post-adjustment of the snowmelt coefficient using SCA information, can still achieve acceptable performance without the need for recalibrating the full model.

This manuscript fits in this state of the art and confirms most of the conclusions above. To me, the most interesting points here are the use of high resolution satellite data and the inclusion of a variety of catchments, with different snow climatologies and various hydrologic characteristics.

Thanks for appreciating the work we have done in this direction. Using high resolution satellite data and a variety of catchments, we believe we have provided robust evidence for our claims about independent, ex post calibration of snowmelt coefficients.

In the revised manuscript, I would invest more effort in leveraging this variety of cathments as a way to draw process-based conclusions from this study that could allow for generalization: how are these catchments representative of specific snow climates? What do differences in results across these catchments tell us in terms of hydrological processes and the applicability of this approach in ungauged regions?

We thank Dr Avanzi for raising this key point. Our results indicate a dependency of model performance on catchment characteristics. As also discussed in our response to Reviewer 1, we observe differences in performance across basins depending on the climate/physiographic features. For the sake of brevity, we present here the results in terms of BIAS and RMSE for SCF derived from EO-Cm,2 (Figures 1 and 2 below). These performance metrics are evaluated against selected physiographic features (mean elevation, forest coverage, and slope) and climatic features (mean precipitation, temperature, and snowfall). Additionally, we distinguish glacierized and non-glacierized basins using different colours.

[Figure]

*Figure 1 Performances in terms of SCF BIAS versus different physiographic and climatic features when using EO-Cm,2.*

[Figure]

*Figure 2 Performances in terms of SCF RMSE versus different physiographic and climatic features when using EO-Cm,2.*

Our analysis reveals a trend of higher errors in lower-elevation and flatter catchments, and a similar increase in error with higher forest coverage. Regarding the climatic features, an inverse relationship with RMSE is observed: basins with higher precipitation and snowfall tend to exhibit lower errors. This is consistent with expectations, as lower precipitation—particularly solid precipitation—typically leads to more ephemeral snow cover, resulting in a greater proportion of fractional snow-covered areas, which are

more prone to detection and modeling errors. Glacierized catchments do not appear to differ substantially in performance compared to non-glacierized ones. One notable exception is the Umealven basin, which consistently appears as an outlier with significantly lower RMSE. This may be due to its prolonged and near-complete snow cover throughout the season, which likely reduces snow cover variability and modeling errors. However, we acknowledge that it remains difficult to draw direct associations between specific climatic conditions and performance outcomes.

Regarding the applicability of the method to ungauged basins, note that both the Adige and Guadalfeo basins are modeled using a regionalization approach. Despite this, our results show that LISFLOOD performs acceptably in terms of SCF for these basins. The improvements introduced by the new snowmelt coefficient (SMC) are comparable to those observed in the gauged basins, suggesting the method's robustness. However, as discussed in the original manuscript, the Guadalfeo basin shows significant underestimation of river discharge. This poor performance could be partially attributed to the limitations of parameter regionalization and/or model assumptions related to reservoir operations. Specifically, the Rules reservoir, which was opened in 2004, is included in the model throughout the entire simulation period, potentially introducing structural inconsistencies. While a more detailed assessment would require further investigation, we believe the proposed method is applicable to both gauged and ungauged basins. The improvements in snow representation appear to provide benefits in either case, particularly in enhancing the accuracy of the snow component without requiring full model recalibration.

Authors also consider a fairly long period of data, with several snow drought episodes. Maybe commenting results across extremes and average years could be another way of bringing about more novelty.

We thank Dr Avanzi for this important suggestion. As shown in the trends of SCA and SWE across Figures 4, 5, and 6 in the original manuscript, snow drought events appear to be reasonably well reproduced by both the original and the revised SMC—for example in the case of the Adige basin during the 2021/22 and 2022/23 seasons. However, in our view, a more detailed evaluation of snow drought representation would require the use of a temporally (or at least seasonally) varying SMC. This is because different characteristics that define snow droughts—such as thinner snowpacks, earlier onset of melt, and accelerated melt rates—are likely tied to different melt dynamics and would benefit from a seasonally adaptive calibration approach. In this study, we opted for a single optimized SMC averaged over five seasons, regardless of whether individual years experienced drought conditions or not. While a detailed exploration of snow drought processes is beyond the scope of this work, we agree that including some of these considerations could add value to the revised manuscript.

The Introduction and the Discussion section could be revised to (1) make the scope of the manuscript broader and (2) discuss how this methodology compares to previous attempts in this realm (see above).

We thank Dr. Avanzi for the comment. We will revise the Introduction according to this and other reviewers' comments.

I agree with authors that a calibration in terms of snow cover fraction is currently the only feasible approach at these scales, even though this requires the additional complication of a SCF parametrization to convert modelled SWE into modelled SCF. This is well discussed in the manuscript, with the only recommendation of providing some additional results on the calibration of the k constants.

We thank the Reviewer for this comment. As described in the manuscript and following the approach suggested by Swenson et al. (2012), $k\_accum$ can be estimated by analyzing $SCF$ and $\Delta$SWE during at the time of the first precipitation event over an initially snow-free pixel. This is done by inverting Eq. 8 of the manuscript and by replacing with $SCF^n=0$. In line with our approach for estimating Cm, we computed kaccum at the pixel level for each of the five "calibration" seasons. The values were bounded to a maximum of 0.5, and pixels for which the coefficient could not be determined were assigned a default value of 0.1. The resulting seasonal values were then averaged over time to obtain a single representative constant.

For the sake of brevity, we did not include these results in the current manuscript, but we present them in Figure 3 below. In the revised version, we might consider including these results in a new appendix to provide additional transparency and methodological detail.

[Figure]

[Figure]

[Figure]

*Figure 3 kaccum for the considered basins.*

In general, the manuscript is well written.

We thank Dr. Avanzi for his valuable feedbacks.

---

## Author Comment (AC3)

Review paper egusphere-2025-2157 (Anonymous Reviewer 3)

**Assessing the Impact of Earth Observation Data-Driven Calibration of the Melting Coefficient on the LISFLOOD Snow Module**

By Premier et al.

We thank the Anonymous Reviewer for providing valuable feedback. We believe that the manuscript can be improved by considering his/her suggestions and by clarifying several critical aspects that were not previously well explained. Below, we provide our point-by-point responses, highlighted in red.

Premier et al. present a calibration method for simulating snow melt using the LISFLOOD hydrological model. The authors tested the method on a sufficient number of basins in Europe and demonstrated the benefits for SCF, SWE, and to a lesser extent, for water balance components.

In my view, the paper's novelty lies in the use of improved high-resolution snow cover data, combined with an SWE to SCF approach and an optimization algorithm. It seems to me that the topic is relevant and of interest to the journal, but it requires a major revision.

We thank the Reviewer for recognizing the importance of the topic and the novelty of the work. We believe that the manuscript can highly benefit from the provided feedback.

The paper is missing some points:

It appears that the authors have introduced the SCF calibration as an alternative to the discharge calibration. Stepwise calibration or data assimilation using snow data, soil moisture, and evapotranspiration has been performed quite often in the last decade. Even with the Lisflood model attempts have been made by Thirel et al. (https://www.mdpi.com/2072-4292/5/11/5825, https://www.sciencedirect.com/science/article/pii/S0034425712003604).

We acknowledge that the novelty of this work does not lie in the use of stepwise calibration or data assimilation techniques, and we do not claim otherwise. We agree with the Reviewer that this point should be stated more explicitly in the revised version of the manuscript. However, as also noted by the Reviewer, to our knowledge these approaches have not previously been applied using fully gap-filled SCA data derived from high-resolution satellite imagery (at least, for LISFLOOD). We believe that the use of such a dataset represents a significant added value.

The paper does not employ a multistep approach (first snow, second discharge) to improve overall calibration. As mentioned, Lisflood is the driving hydrological model of EFAS and GloFAS, which focus on discharge forecasting. An improvement in SCF is fine, but it cannot be the final goal. An improvement in SCF can even lead to a worse

objective criterion of discharge, but might still be an improvement because it reduces the error of overfitting.

We agree that a multistep calibration approach would be a final goal for the full calibration of the model. This study focuses on the first step and the downstream consequences of calibrating only the SMC without modifying the other parameters. Existing models typically generate SCA maps that are broadly correct but often lack details. We introduce a method to improve the fine-scale representation of these SCA maps. A key aspect of our work is that these improvements are achieved without affecting the mean SCA, which in turn preserves the discharge accuracy.

This is the reason why the analysed catchments do not show a significant decrease in KGE when the model was run with the new SMC, so in case of a new calibration we expect the equivalent KGE or an improved one. We will expand on this in the last comment.

However, given the importance of the 2 system at global and European level, a 2-step approach would have to go through an increased number on tests, and a full 2-step calibration approach, since the results in KGE could be different in other areas, as suggested by the reviewer.

Here, the focus is solely on the snow ablation process, using the snow melt coefficient (SMC) as the parameter. The process of snow accumulation, with parameters such as snow factor or temperature threshold that determine whether precipitation falls as rain or snow, is overlooked.

We thank the Reviewer for this insightful comment. We agree that our current focus on the snow ablation process, represented by the snow melt coefficient (SMC), overlooks important aspects of the snow accumulation phase. As acknowledged in lines 244–246, we did not explicitly address parameters such as the temperature threshold that determines whether precipitation falls as rain or snow (e.g., $Tm$, usually set around 1 °C), or the snow factor.

We fully recognize the importance of accurately representing and tuning the snow accumulation process. In fact, errors in this phase may significantly impact the overall snowpack evolution. While such tuning can indeed be achieved by adjusting model parameters, we believe a major source of uncertainty may lie in the precipitation input data itself and specifically, in its amount, leading to incorrect accumulation regardless of model settings.

Additionally, we believe that relying solely on snow cover fraction (SCF) data may not be sufficient for constraining the accumulation process, particularly since precipitation events can occur over already snow-covered areas, where SCF remains unchanged.

Therefore, while SCF is valuable, it may not provide enough information to fully and accurately calibrate accumulation-related parameters.

However, we believe that considering an average SMC over different hydrological years is also smoothing the effects of possible errors in precipitations. In any case, we acknowledge this as a limitation of the current study and plan to explore the accumulation process more thoroughly in future work.

The Lisflood snow modul is not explained fully. It is mentioned that Lisflood uses three different elevation zones (line 134), but it is not explained how the SCF calculation from SWE, the calculation of SMC (e.g., equation 12), or the optimization is performed with these three zones.

Thanks for the comment. We improved the LISFLOOD documentation (https://github.com/ec-jrc/lisflood-model/blob/jcr_revision/2_04_stdLISFLOOD_snowmelt/index.md) that will be published when LISFLOOD version 5.0 will be released. However, we did not consider the three elevation zones in the SCF parametrization, Eq. 12 or for the optimization approach. The three elevation zones play a role only in the SWE calculation, and in more detail in the temperature and consequently in the accumulation and ablation processes. Afterwards, a single SWE value per pixel is considered and the rest of the approach refer to the "average" SWE value for each pixel.

Especially with higher resolution (here 1 arcmin) the day-degree approach can accumulate too much snow at high altitudes, as temperatures will not too often drop below 1° C. Lisflood uses a workaround to melt additional snow in Summer (IceMeltS). The paper does not mention this approach, nor does it take it into account.

Thank you for the comment. To partially address the issue that the degree-day approach can lead to excessive snow accumulation at higher elevations—especially when applied at higher spatial resolutions (such as 1 arcmin)—the use of three elevation zones was introduced. This zonal approach helps to mitigate overestimation of snow accumulation by better accounting for the altitudinal variation in temperature and snow dynamics. Regarding your comment on the Ice Melt integration, this contribution was neglected in this work. However, we can include it for completeness in the new results.

The effect on the water balance is calculated on a monthly basis, even when the model is run on 6-hour timesteps. Here, it is really necessary to go on daily basis. With monthly evaluations, you miss the main advantage of your SCF calibration: having a better estimate of the timing of the main snow ablation, and therefore a more accurate estimate of the timing and magnitude of spring floods.

We thank the reviewer for the comment. We will remove the monthly analysis and show the daily analysis (as shown in the answer to the comments to Reviewer 1 and the

answer and figures below). We agree that that daily analysis is showing better differences in the timing of the peak. We will also include the calculation of a seasonal KGE, showing the KGE and the relative metrics per season.  Our analysis has highlighted negligible differences in terms of the overall metrics, however calculating metrics on a seasonally base might highlight a worse/better behaviour in the timing and magnitude of spring/summer floods.

What we regard as an important practical implication for modelling, though, is exactly the possibility of calibrating the SMC directly on snow cover and run the model with the other parameters unchanged, without deteriorating the performance on the streamflow. It is therefore possible to create a setup of the model arguably more realistic in terms of snow cover, without the need to recalibrate the other parameters. Not only is calibration of large models resource-intensive, but there is also a risk of model overfitting that is constrained according to our procedure. See also reply to the last specific comment by the reviewer below.

Plots with daily averages are attached to the document (Figure 4).

Discharge wise, Salzach river shows a worse performance, with lower daily discharge between February and April, while in June and July the discharge is overestimated, also performing worse than the benchmark.

The Arve River shows a better performance in July and beginning of August, with a good match against observed data, worse performance in February and March.

The Alpenrhein has an improved performance in June-July, but worse in March-April.

In the other catchments differences are negligible.

To investigate whether this is associated with poorer performance in terms of SCF, we further analyzed the trends of bias and RMSE across the different months. The results are presented in the figures below. Notably, worse SCF performance is observed particularly in April for the Salzach, and in March and April for the Arve. The Alpenrhein

shows a higher bias in April, along with higher RMSE values from April through June.

[Figure]

*Figure 1 Monthly error trends in terms of SCF for the Salzach basin.*

[Figure]

*Figure 2 Monthly error trends in terms of SCF for the Arve basin.*

[Figure]

*Figure 3 Monthly error trends in terms of SCF for the Alpenrhein basin.*

The 2nd step of calibration for discharge is missing, as is an explanation of how to derive a better KGE with a change in SMC. In L_Cm version of Lisflood, SMC is calibrated to

improve the KGE (SMC is optimized for discharge KGE). In the EO-Cm version, you changed only the SMC, and you keep all the other calibration parameters? The improvement in KGE (even the tiny one) can only be explained by a bigger range of SMC and/or by the single cell values. However, the calibration was performed on daily discharge; therefore, a comparison with daily values would be appropriate.

A 2nd discharge calibration is necessary to see the improvement vs the original calibration, using the new SMC as predefined values. I am not asking for all 9 basins but for those where you have only one subbasin (Arve, Laborec, Morrumsan, Umealven)

In the EO-Cm version, we changed only the SMC keeping all the other calibration parameters the same. It is true that the bigger range of SMC can be an explanation for the improved KGE, we will include that in the discussion and highlight the differences between the 2 snowmelt coefficients.  We addressed the calibration comment in the specific comments below.

**Specific comments:**

- L2: I would not call it traditionally. It is not made because of tradition, but it has a reason. If you call it later traditional calibration, it is ok

  Thanks for pointing this out. We removed it.

- L21: This is unclear. It cannot be globally between 40-90% snow contribution from mountains. Please check Viviroli again

  Thank you for the comment. Our intention was to refer to *regional* contributions rather than *global* ones. This is clarified in the Introduction of Viviroli et al. (2007), which states: "regionally, mountain discharge may represent up to 95 percent of total flow in a catchment [Liniger et al., 1998]." This is further supported by Viviroli et al. (2004): "In humid areas, mountains supply up to 20–50% of total discharge, while in arid areas, mountains contribute from 50–90%, with extremes of over 95%."

  To avoid confusion, we will revise the sentence in the manuscript to read:
  "Depending on geography and climate, the regional contribution of snow to river runoff can vary substantially, from as low as 40% up to 95% of the total annual flow (Viviroli et al, 2007)."

- L25: "LISFLOOD is one of the most comprehensive operational models used in Europe to simulate hydrological processes". This is very general sentence. Maybe a unique selling point: Lisflood is one of few operational models calibrated for Europe to simulate hydrological processes.

Thanks. We changed with "LISFLOOD is one of the few operational models used in Europe to simulate hydrological processes."

- L34: I think the equation which takes rain into account is from: Speers, D.D., Versteeg, J.D. (1979) Runoff forecasting for reservoir operations - the past and the future. In: Proceedings 52nd Western Snow Conference, 149-156

We revised the sentence: "The snow module within LISFLOOD simulates snowmelt using a temperature-based approach—specifically, a degree-day model that also accounts for enhanced snowmelt during rainfall events (Speers and Versteeg,1982)."

- L65: for a "novel" method you explain not much in L183-184

Thanks for the comment. As explained later, the optimization technique itself is not novel and it is part of the SciPy library. What we meant, is that this optimization approach has not been previously applied to LISFLOOD. For the sake of transparency, we will use the word "alternative" instead of "novel".

- L103: "The current model setup operates …". Maybe put this after line 106, because the first part explains Lisflood, the second a special application of Lisflood for the EFAS setting

We inverted the two sentences.

- L131 it is rainfall per day not rainfall intensity. Somewhere else it should be hydrological year instead hydrological season

Thanks. We replaced with the correct terminology.

- L136: The 3 zones can be explained in more detail and has to be included in 2.3 and 2.4.

We thank the Reviewer for the comment. Also as pointed out by Reviewer 1, our explanation of how elevation zones were defined was not sufficiently clear. Due to the relatively coarse resolution of LISFLOOD cells (1', ~1.4 km), significant sub-pixel variability in snow accumulation and melt can occur, particularly in areas with large elevation differences within a single pixel.

To address this, snow processes are modeled separately within three elevation zones defined at the sub-pixel level. These zones are determined based on a normal distribution of elevation values, which has been shown to represent well the actual distribution. To this purpose, the standard deviation of elevation within a grid cell is calculated from the Multi-Error-Removed Improved-Terrain (MERIT) DEM with a spatial resolution of 90 m. The three elevation zones—A, B, and C—are each assumed to cover one-third of the pixel area.

Assuming that the average pixel temperature corresponds to the mean pixel elevation, temperatures for the lower zone A and upper zone C zones are estimated by applying a fixed lapse rate (L = 0.0065 °C/m) to the elevation differences from the mean. Snow accumulation and melt are then modeled separately for each zone, using the temperature at each zone's centroid as a proxy for local conditions.

To improve clarity, we will add these details that can be found in the LISFLOOD model official documentation (https://github.com/ec-jrc/lisflood-model/blob/jcr_revision/2_04_stdLISFLOOD_snowmelt/index.md) to the manuscript, that has been recently revised to improve clarity.

- Also the IceMelt part in https://github.com/ec-jrc/lisflood-code/ is not explained at all and not taken into account.

Thanks for the comment. We revised the documentation regarding the ice melt part at https://github.com/ec-jrc/lisflood-model/blob/jcr_revision/2_04_stdLISFLOOD_snowmelt/index.md, that will be published as soon as LISFLOOD version 5.0 is released.

At high altitudes, where the temperature never exceeds 1°C, the model accumulates snow as the temperature threshold for melting (Tmelt) is never exceeded. In these altitudes runoff from glacier melt is an important part. Snow will accumulate and convert into firn; then, firn is converted into ice and transported to the lower regions. This process can take decades or even hundreds of years. In the ablation area the ice is melted.

In LISFLOOD, this process is emulated by melting the ice in higher altitudes on an annual basis over summer.

$$IM_z = T_z \cdot C_{im} \cdot \Delta t$$

where:

- IMz is the icemelt (mm) per time step and elevation zone.
- Cim is the seasonally-varying icemelt coefficent (mm/°C day).

The seasonal icemelt coefficient enforces that icemelt only happens during summer (from June 13 to September 13 in the Northern Hemisphere, from December 13 to March 14 in the Southern Hemisphere). It also takes the shape of a sine function with a maximum value of 7mm/°C day:

$$C_{im} = \begin{cases} 7 \cdot \sin\left( (\text{doy} - \text{start}) \cdot \dfrac{\pi}{365.25} \right) & \text{if start} < \text{doy} < \text{end} \\ 0 & \text{else} \end{cases}$$

where start and end are the days of the year representing the beginning and end of the icemelting season, i.e., approximately summer.

However, in our exercise we did not include this component since the glacier-covered area is relatively small in most of the glacierized basins—less than 1% of the total area (approximately 0.9% for Adige and Salzach, and 0.6% for Alpenrhein)—we acknowledge that glaciers can still have a non-negligible influence, particularly in the Arve basin, where the glacierized area is approximately 5%. Our initial intention was to mask out pixels where the glacier coverage exceeded a certain threshold during the calibration of the snowmelt coefficient. A proper representation of glaciated areas would require distinguishing between snow and ice surfaces and applying different coefficients accordingly. However, we believe this is beyond the scope of our work. The simplified approach as implemented in LISFLOOD can be integrated in the next version.

- L146ff: This part can be done in a nicer way. E.g. https://egusphere.copernicus.org/preprints/2025/egusphere-2025-1214/egusphere-2025-1214.pdf has a better way to structure this.

Thank you for the comment. We have reviewed the suggested paper, but we are not entirely certain about the specific concern raised—whether it relates primarily to stylistic presentation (e.g., adding a table with symbol definitions) or to conceptual clarity. Nevertheless, we will revise the section to improve clarity, and we will consider including a table of symbols if it enhances readability. These revisions will also be aligned with the changes made in response to Reviewer 1's comments.

- L152: How is kaccum calculated?

We thank the Reviewer for this comment. As described in the manuscript and following the approach suggested by Swenson et al. (2012), $k\_accum$ can be estimated by analyzing $SCF$ and $\Delta SWE$ during at the time of the first precipitation event over an initially snow-free pixel. This is done by inverting Eq. 8 of the manuscript and by replacing with $SCF^n=0$. In line with our approach for estimating Cm, we computed $k\_accum$ at the pixel level for each of the five "calibration" seasons and then averaged these values over time to obtain a representative constant. We will explain this better in the revised manuscript. An

answer and maps representing the used $k\_accum$ coefficients are provided also to Reviewer 2.

- L155: What is the reference of this equation?

All this part refers to the methodology proposed by Swenson and Lawrance (2012). More in detail, for the equations please refer to the code of the CLM model, available here: CTSM/src/biogeophys/SnowCoverFractionSwensonLawrence2012Mod.F90 at master · ESCOMP/CTSM

With respect to the original paper, there are some differences. In detail, Equation 11 is reported with a typographical error; however, the correct formulation is implemented in the corresponding code. We have also been in contact with the original authors to confirm that we are using the correct version of the formula. Anyway, as also reported to Reviewer 1, we plan to make this part clearer in the next version.

- L161: How is this calculated with the 3 elevation zones

Thanks for the comment. We have explained in a previous answer how the elevation zones play a role in the computation of a different temperature. These zones are determined based on a normal distribution of elevation values, which has been shown to represent well the actual distribution. To this purpose, the standard deviation of elevation within a grid cell is calculated from the Multi-Error-Removed Improved-Terrain (MERIT) DEM with a spatial resolution of 90 m. The three elevation zones—A, B, and C—are each assumed to cover one-third of the pixel area. However, this does not affect the SCF parametrization, as explained in a previous answer.

- L169f: The equation 11 is invalid for glaciers and cannot applied for areas with always snow and with several snow-cycles. Equa 12 is again without the snow elevation zones.You showed it anyway, that this equation is not leading somewhere. It is fine to keep this approach.

Thanks for the comment. We agree that this equation is a simplification, and it is not valid in the particular cases mentioned by the Reviewer (as mentioned in L173). Anyway, both this Equation as well as the optimization approach, are not applied per each elevation zone. As stated in L136, the processes that are modeled separately for 3 separate elevation zones to take into account sub-pixel heterogeneity linked to elevation differences given the large pixel size are only the snow melt and accumulation.

- L183f: You featured this as "novel" approach. It appears to be an optimization for a single-parameter part of the standard SciPy library. Does seem to be novel and not explained at all.

  Thanks for the comment. You are absolutely right, the optimization technique itself is not novel at all and it is part of the SciPy library. What we meant, is that this optimization approach has not been previously applied to LISFLOOD and specifically to compute Cm, while previous work as the one of Pistocchi et al., 2017 has focused on simpler methodologies. On the other hand, other works as Thirel et al., 2012 or Thirel et al., 2013 have focused on a comparison or data assimilation. We will revise the sentence to make this clearer.

- Table1: the max elevation is not explained. In the original Merrit DEM it is much higher, so I assume you average that for 1 arcmin. But in the model the max elevation is the highest elevation zone. I think you should correct the elevation by the values from original Merrit DEM
  Thanks for the comment. Yes, you are absolutely right that this can be misleading and depending on the resolution of the used DEM, the values can be different. The DEM (also showed in Figure 1 of the manuscript) is the one aggregated at the LISFLOOD resolution of 1 arcmin. Therefore, we will retain the figure and the color bar ranges as they currently are, as they correspond to the 1 arcmin DEM. Hence, we will clarify this in the figure caption to avoid confusion by adding "The elevations reported here (and hence the color bar ranges) refer to the MERIT DEM aggregated at the model resolution of 1 arcmin."

  Regarding Table 1, we consider it more appropriate to report the elevation values that correspond to the minimum and maximum heights used in the model, based on the division into three elevation zones. Therefore, we define the minimum elevation as **min(DEM$_i$ – σ$_i$)** and the maximum as **max(DEM$_i$ + σ$_i$)**, reflecting the lower and upper bounds of the elevation ranges we model. We will update both the table and its caption accordingly to clarify this approach.

**Table 1.** Overview of the nine hydrological catchments selected in this study, including their respective countries, area, and elevation information. The mean elevations reported here refer to the elevation from the MERIT DEM aggregated at the model resolution of 1 arcmin. Maximum and minimum elevations are calculated considering $\sigma_{topo}$, thus representing the minimum and maximum elevations modelled.

| Basin | Countries | Area [km$^2$] | Elevation [m] | | |
|---|---|---|---|---|---|
| | | | min | mean | max |
| Adige | Italy | 12100 | 3 | 1497 | 3724 |
| Alpenrhein | Switzerland/Austria | 7400 | 389 | 1660 | 3276 |
| Arve | France | 2000 | 344 | 1372 | 4413 |
| Gállego | Spain | 3900 | 199 | 798 | 2922 |
| Guadalfeo | Spain | 1200 | 70 | 1293 | 3294 |
| Laborec | Slovakia | 1300 | 136 | 421 | 980 |
| Mörrumsån | Sweden | 3500 | 9 | 186 | 310 |
| Salzach | Germany/Austria | 6700 | 348 | 1260 | 3295 |
| Umeälven | Sweden | 6100 | 318 | 711 | 1557 |

- L297ff: Why using a monthly comparison. The original model is calibrated on daily values. The biggest effect of your improvents are the timing in days of the biggest drop in snow accumulation. The comparison to daily discharge is necessary to conclude if the models performance is improved (maybe the over discharge KGE is reduced, but some spring floods are better timed, the snow cover is better estimated).

We thank the reviewer for the comment we will include a comparison on average daily discharge over years, as shown in the following figures.

[Figure]

[Figure]

[Figure]

*Figure 4 Comparison of daily average discharge over years for the different catchments. In cyan, the discharge computed with the old coefficient, in orange the discharge with the new coefficient and in black the observed discharge.*

We made some preliminary analysis on peak timing of discharge; however, it was difficult to have drawn conclusions looking at single events. In some cases, peak was better captured in some other events no, in other events no difference was noticeable etc., this can be explained with the fact that are other factors contributing to the discharge at the location of the observed river discharge. The results are shown in the following figures.

[Figure]

*Figure 5 Peak events for the Adige basin.*

[Figure]

Figure 6 Peak events for the Alpenrhein basin.

[Figure]

Figure 7 Peak events for the Arve basin.

[Figure]

Figure 8 Peak events for the Gallego basin.

[Figure]

Figure 9 Peak events for the Guadalfeo basin.

[Figure]

*Figure 10 Peak events for the Laborec basin.*

[Figure]

*Figure 11 Peak events for the Morrumsan basin.*

[Figure]

*Figure 12 Peak events for the Salzach basin.*

[Figure]

*Figure 13 Peak events for the Umealven basin.*

Another possible analysis that can be included in the revised version of the manuscript is to calculate KGE and its metrics grouped by season/month and see if we see improvements or worsening in performance.

- Fig 7: Gallego and Guadelfeo have some reservoirs included. It would be better to use subbasins without too much human interference. From the results, you cannot see if it is the snow or the reservoirs. You explained why Guadalfeo has a bad KGE. One solution could be to use only those years without reservoirs,

The observed discharge for Guadalfeo river starts from 2014, so the bad performance is due the regionalized parameterization, we will specify that in the revised version of the manuscript.

We agree with the reviewer that ideally the less human influenced the better, however LISFLOOD is calibrated on human influenced streamflow we wanted to include catchments with different characteristics.

In the case of the Gallego river, the reservoir (highlighted in orange in Figure) might influence partially a change in discharge as shown in the Figure 14 below.

[Figure]

[Figure]

*Figure 14 Normalized Euclidean Distance between daily discharge of LISFLOOD model as per in EFAS 5.0 and the LISFLOOD model run with the new snow melt coefficient. Light colours mean that the discharge are different, dark colors mean that discharges are similar. Reservoirs are highlighted in orange in the figure.*

- Table 4: This is not suitable for comparison. 1) you keep the other parameter constant (I assume, it is missing in the paper) 2) you compare on monthly values 3) you did not recalibrated the other parameters after setting SMC to your values.

  Thanks for the insightful comment.

  1) Correct. We will highlight that better in the manuscript.
  2) No metrics are calculated on daily values of river discharge
  3) Correct.

- I think it is necessary to re-calibrate for a number of basins (maybe only those where you do not have upstream-downstream calibration) and discuss the effect of your improved SMC e.g. worse KGE but better representation of snow, more exact timing of snow-induced flooding,

  We thank the reviewer for the useful comment. We compared the simulation of streamflow with the new and old SMC with all other parameters unchanged, and the differences are most of the times very small or even negligible, the biggest difference in KGE takes place in the Laborec catchment (difference = 0.05)

Given the fact that the calibration maximises the objective function (KGE), by recalibrating the model we do not expect a lower KGE compared to the one obtained by running LISFLOOD with the new SMC and the current LISFLOOD parameters.

So, this reinforces three arguments that we will stress better in the revised manuscript:

- A possible 2 step calibration (1. Snowmelt coefficient on EO 2. remaining parameters), has the potential to improve the KGE. In the catchments we analysed, the KGE was slightly degraded in some cases, but within a narrow margin.
- This procedure allows the integration of more realism on the snow component of the LISFLOOD model without recalibrating the other parameters. This may be very useful when in need to use this large-scale model for specific purposes in snow-dominated catchments, while preserving consistency with the overall dynamics at the larger scale.
- The LISFLOOD model as currently calibrated and implemented for EFAS 5, can capture the average snow dynamics at the calibration station. Snowmelt dynamics are likely to change (as shown in the Euclidean distance figures) upstream, so river discharge affected by snowmelt dynamics should be treated with cautions upstream the calibrated stations.

Overall, the topic is interesting, and the potential for a good paper is there, but it lacks structure, and fundamental key points are not included yet.

We believe the responses to comments above have cleared the way to improving the manuscript as required by the reviewer.